



# Rate coefficients for reactions of OH with aromatic and aliphatic volatile organic compounds determined by the Multivariate Relative Rate Technique

Jacob T. Shaw[1*], Andrew R. Rickard[1,2], Mike J. Newland[1] and Terry J. Dillon[1]

5   [1] Wolfson Atmospheric Chemistry Laboratories, Department of Chemistry, University of York, Heslington, York, YO10 5DD, UK

[2] National Centre for Atmospheric Science, University of York, Heslington, York, YO10 5DD, UK

* Now at Department of Earth and Environmental Science, University of Manchester, Manchester, M13 9PL, UK

*Correspondence to*: Terry J. Dillon (terry.dillon@york.ac.uk)





**Abstract.** The multivariate relative rate method was applied to a range of volatile organic compounds (VOC) reactions with OH. This previously published method (Shaw et al., 2018b) was improved to increase the sensitivity towards slower reacting VOC, broadening the range of compounds which can be examined. A total of thirty-five room temperature relative rate coefficients were determined; eight of which have not previously been reported. Five of the new reaction rate coefficients were for large alkyl substituted monoaromatic species recently identified in urban air masses, likely with large ozone production potentials. The new results (with $k_{OH}$ (296 K) values in units of $10^{-12}$ cm$^3$ molecule$^{-1}$ s$^{-1}$) were: *n*-butylbenzene, 11 ($\pm$ 4); *n*-pentylbenzene, 7 ($\pm$ 2); 1,2-diethylbenzene, 14 ($\pm$ 4); 1,3-diethylbenzene, 22 ($\pm$ 4) and 1,4-diethylbenzene, 16 ($\pm$ 4). Interestingly, whilst results for smaller VOC agreed well with available structure activity relationship (SAR) calculations, the larger alkyl benzenes were found to be less reactive than the SAR prediction, indicating that our understanding of the oxidation chemistry of these compounds is still limited. $k_{OH}$ (296 K) rate coefficients (in units of $10^{-12}$ cm$^3$ molecule$^{-1}$ s$^{-1}$) for reactions of three large alkanes with OH were also determined for the first time: 2-methylheptane, 9.1 ($\pm$ 0.3); 2-methylnonane, 11.0 ($\pm$ 0.3) and ethylcyclohexane, 14.4 ($\pm$ 0.3), all in reasonable agreement with SAR predictions. Rate coefficients for the twenty-seven previously studied OH + VOC reactions agreed well with available literature values, lending confidence to the application of this method for the rapid and efficient simultaneous study of gas-phase reaction kinetics.



## 1 Introduction

The troposphere contains many thousands of organic compounds (Lewis et al., 2000; Goldstein and Galbally, 2007). These compounds differ substantially in their chemical and physical properties. Volatile organic compounds (VOC) are organic chemicals which take part in photochemical degradation cycles leading to the formation of ground-level ozone and photochemical smog, and the production of secondary organic aerosols – significantly impacting upon air quality and climate.

Aromatic VOC can account for a large proportion of the non-methane hydrocarbon mass in the atmosphere. At least ninety different aromatic compounds have been identified in diesel fuel (Hamilton and Lewis, 2003). Aromatics are therefore particularly prevalent in heavily urbanised environments associated with large amounts of fuel combustion and solvent use. For example, aromatic VOCs were reported to account for up to 33% of the VOC emission mass from vehicle exhausts in China (Guo et al., 2011; Cao et al., 2016). The total concentration of five aromatic compounds was measured to be 27 ($\pm$ 8) ppbv in Mumbai (Pandit et al., 2011) while summed concentrations of eleven aromatic compounds were measured to be 30 ppbv in Yokohama City (Tiwari et al., 2010). Peak concentrations of aromatic VOCs can be much greater, however; Dunmore et al. (2015) measured daily winter-time peaks for $C_4$ substituted monoaromatics of 500 ppbv, associated with extensive traffic pollution in London.

Whilst many aromatic species are known to exist in the atmosphere, measurements of OH + aromatic VOC rate coefficients are sparse. The IUPAC Task Group on Atmospheric Chemical Kinetic Data Evaluation provides evaluated rate coefficients for twenty-six OH + aromatic reactions, including cresols, phenols, nitrophenols, quinones and furans (Atkinson et al., 2006). Calvert et al. (2002) provides an evaluation of a further eighteen reactions between OH and aromatic VOCs with alkyl substituents. More recently, Jenkin et al. (2018a) have developed an updated aromatic + OH structure-activity relationship (SAR) based on a compiled database of sixty-seven monocyclic aromatic hydrocarbons and oxygenated organic compounds, including eight higher alkyl-substituted benzenes and four alkenylbenzenes. Despite their ubiquity in the urban atmosphere, repeat measurements of OH + aromatic VOC rate coefficients by different laboratories are rare; only toluene, *o*- and *m*-xylene and the three isomers of trimethylbenzene have five or more example measurements in the literature. The number of available rate coefficient measurements drastically decreases with increasing substituent size of aromatic VOCs.

The propensity for VOCs to generate tropospheric ozone through their photochemical oxidation is indicated by their photochemical ozone creation potentials (POCP). This model calculated reactivity index describes the relative capability for an organic compound to produce ozone over north-west Europe (Derwent et al., 1998, 2007). Di- and tri-substituted $C_1$, $C_2$ and $C_3$ aromatics have relatively large POCP values meaning that they are amongst the emitted species with the greatest potential for producing ozone (Derwent et al., 2007).

Many measurement techniques fail to detect aromatic compounds with larger alkyl substituents (greater than three carbons) and are therefore limited to quantifying the smaller substituted aromatics, such as toluene, xylenes, and trimethylbenzenes. In the absence of measured values, toluene-equivalency ratios can provide a prediction of the additional aromatic content of the atmosphere based on measured concentrations of the smaller aromatic compounds (Lidster et al., 2014).




However, using toluene concentration as a proxy for concentrations of larger aromatics requires knowledge of their rate coefficient for reaction with OH; this is often unavailable for $C_4$ and larger aromatics.

Reactions between aromatic VOCs and the hydroxyl radical (OH) primarily involve OH addition to the aromatic ring. For benzene, and the methyl- and ethyl-substituted aromatics this pathway dominates at room temperature, accounting for greater than 90% of the OH reaction. The reaction produces a hydroxyalkylcyclohexadienyl radical, or OH-aromatic adduct, which can decompose back to VOC + OH, or be captured by reaction with $O_2$ or $NO_2$. Pseudo first-order rate coefficients for the OH-adduct + $O_2$ capture reaction and for the OH-adduct back decomposition are 1000 s$^{-1}$ and 1 s$^{-1}$ respectively at [$O_2$] = 0.2 atm (Vereecken et al., 2019). Thus, while in the troposphere, effectively all of the OH-adduct is captured by reaction with $O_2$. In laboratory studies performed at low $O_2$ concentrations, the competition of these pathways can lead to non-atmospherically relevant product distribution and must be considered (Ji et al., 2017; Newland et al., 2017). The OH-adduct can also react with $NO_2$, but the ratio of the $O_2$ and $NO_2$ rate coefficients is such that, under typical atmospheric conditions, monocyclic aromatics react almost exclusively with $O_2$ (Knispel et al., 1990; Koch et al., 1994; Bohn and Zetzsch, 1999; Bohn, 2001). Meta-substituted aromatics, such as *m*-xylene, react faster than their ortho- and para- counterparts owing to enhanced electrophilic addition of OH leading to the formation of a resonance stabilised tertiary radical addition product (Ravishankara et al., 1978; Nicovich et al., 1981; Mehta et al., 2009). Arrhenius plots for the reactions of VOCs with OH tend to exhibit complex behaviour; two distinct regions are visible over a large temperature range. It is thought that the OH-aromatic adduct undergoes thermal decomposition at temperatures greater than 325 K, so that the H-atom abstraction pathway becomes the dominant reaction mechanism as temperature increases. The atmospheric lifetime of the OH-aromatic adduct is predicted to be approximately 0.3 s at 298 K (Atkinson and Arey, 2003). This decreases rapidly with temperature to an approximate lifetime of 20 μs at 450 K. To gain a thorough understanding of the oxidation of aromatic VOCs, these reactions need to be studied across the typical range of atmospheric conditions.

Alkanes are also ubiquitous in the urban atmosphere and are generally associated with vehicle emissions. Approximately 40% of the chemicals found in exhaust emissions are alkanes (Guo et al., 2011; Cao et al., 2016) but this can increase to up to 90% when the engine loading is above 30% (Pereira et al., 2017). Long-chain diesel related aliphatic hydrocarbon species ($C_{10}$-$C_{13}$) have been observed to dominate the gas phase reactive carbon in cities with high diesel fleet fractions (Dunmore et al., 2015) and have also been detected in engine exhaust (Erickson et al., 2014).

The relative rate method is an established technique for the measurement of rate coefficients for the reactions of VOCs with OH. Recent developments in the analytical capabilities of instrumentation have allowed for the development of an extension to the technique, in which multiple reactions can be monitored simultaneously (Shaw et al., 2018a; Shaw et al., 2018b). Rather than measuring a target reaction relative to a single reference reaction, target reactions are placed on an absolute basis relative to a suite of similar reference reactions, reducing the reliance on using any single value as a reference. This multivariate technique has been used to measure the rates of reaction for nineteen VOC with OH at room temperature, and for twenty-five VOC (twelve of which were aromatic VOC) with OH at an elevated temperature of 323 (± 10) K (Shaw et al., 2018b).





Clearly there remain significant gaps in our understanding of atmospheric oxidation chemistry. Accordingly, in this work, laboratory based experiments were conducted to study the gas-phase kinetics of a suite of atmospherically important reactions of aromatic and aliphatic VOCs with OH. Use of the multivariate relative rate technique allowed the rapid determination of thirty-five such rate coefficients ($k_{OH}$ (296 K)) including for $C_3$ and $C_4$ mono-substituted benzenes, a range of diethylbenzene isomers, and a number of methyl- and ethyl-branched alkanes that were, to the best of our knowledge, previously unknown.





## 2 Methodology

A detailed description of the multivariate relative rate technique was provided in Shaw et al. (2018b). Briefly, three synthetic gas-mixtures were prepared by injecting 1–7 mm³ of undiluted VOC into a 500 cm³ evacuated sample cylinder (Swagelok) and flushing through into a pacified gas-cylinder (10 L, Experis, Air Products) with $N_2$. A two-stage dilution process was required to achieve approximate mixing ratios of each VOC in the cylinder of 40 parts per billion by volume (ppbv). Gas-phase reactants were mixed in a flow reactor under conditions of 1 bar ($N_2$) and 296 ($\pm 2$) K. Photochemical oxidation reactions were initiated through the use of a low-pressure Hg/Ar lamp (L.O.T., Pen-Ray®); OH radicals were generated by the photolysis of $H_2O$ via Reaction (R1). It was expected that the atomic H by-product was rapidly converted to $HO_2$ by reaction with molecular oxygen, which could not be fully excluded from the reactor (R2).

$$H_2O + h\nu \text{ (184.95 nm)} \rightarrow OH + H \tag{R1}$$
$$H + O_2 + M \rightarrow HO_2 + M \tag{R2}$$

To limit photolysis of the aromatic VOC, by either the 185 nm emission or by the principal Hg emission band at 254 nm, the gas mixture was introduced into the reactor downstream of the photolysis zone via a moveable injector (Cryer, 2016; Shaw et al., 2018a). Depletion of VOC due to photolysis was not observed for mixtures containing *m*- and *o*-xylene (Shaw et al., 2018a). Flow rates of the VOC gas mixture and the diluting $N_2$ bath gas containing $H_2O$ were controlled by various mass flow controllers (MKS and Tylan). Assuming a constant total flow rate of 3000 cm³ min⁻¹ into the reactor, the VOC were estimated to have a residence time of 4 s, of which OH was estimated to persist for no longer than 0.5 s.

Samples were collected with the lamp alternately switched off and on, thereby generating two datasets; one in which the VOC concentrations were representative of their cylinder concentration, and one in which the VOC concentrations were depleted by reaction with OH. The depletion factor for a VOC due to reaction with OH ($\ln\left(\frac{[VOC]_0}{[VOC]}\right)$) was related to its rate coefficient for reaction with OH by Eq. (1) (Shaw et al., 2018a). The integral of OH concentration as a factor of time is referred to as the OH exposure ($OH_{exp}$); all VOC in the gas mixture should experience identical exposure to OH radicals. By simultaneously measuring depletion factors for a range of VOC with different reactivities towards OH, the system was calibrated with respect to rate coefficient. A rate coefficient value for the reaction between each VOC and OH could then be calculated relative to all other reactions, by placing the depletion factor vs *k* relationship on an absolute basis.

$$\ln\left(\frac{[VOC]_0}{[VOC]}\right) = k_{VOC+OH} \int [OH]_t dt \tag{1}$$

Producing OH via the photolysis of $H_2O$ also resulted in the production of $HO_2$ (R2). Although $HO_2$ is much less reactive towards VOCs than OH, it may represent a significant OH sink within the reactor via Reaction (R3). The recommended rate



coefficient for this reaction is $k_3 = 1.1 \times 10^{-10}$ cm$^3$ molecule$^{-1}$ s$^{-1}$ (Atkinson et al., 2006); thus this reaction may proceed at a rate fast enough to compete with the VOC for OH.

$$OH + HO_2 \rightarrow H_2O + O_2 \qquad\qquad\qquad\qquad (R3)$$

Whilst this was not an issue with the synthetic gas mixtures measured in Shaw et al. (2018b), some of the mixtures in this work contained slower reacting VOC. Hence, the HO$_2$ may outcompete the VOCs for OH, resulting in only negligible depletions in the VOC concentrations.

The impact of HO$_2$ was minimised by adding NO into the reactive mix. This served two purposes. Firstly, NO reacts

10 with HO$_2$ (R4) with a recommended rate coefficient of $k_4 = 8.5 \times 10^{-12}$ cm$^3$ molecule$^{-1}$ s$^{-1}$ (Atkinson et al., 2006). Although this value is much smaller than that for the HO$_2$ + OH reaction, the addition of significant quantities of NO reduced the amount of OH lost to reaction with HO$_2$. Secondly, one of the products of the HO$_2$ + NO reaction is OH. This reaction therefore served as a secondary source for OH radicals, effectively converting the photolysis by-product (HO$_2$) into useful OH.

15 $$HO_2 + NO \rightarrow OH + NO_2 \qquad\qquad\qquad\qquad (R4)$$

The GC-MS system used for chromatographic separation and detection was an Agilent 6890 (Agilent Technologies) fitted with a DB5-MS ultra-inert capillary column (60 m × 0.32 mm ID × 1 μm film, Agilent Technologies), coupled to a Markes International BenchTOF© mass spectrometer. The temperature ramping in the GC oven varied between mixtures to 20 optimally resolve all mixture components; the sample turnarounds, across both GC method and TDU setup, was approximately 20 min. For a full description of the gas sampling and analysis, please refer to Shaw et al. (2018b).



## 3 Results and discussion

Experimental results from four synthetic gas mixtures containing aromatic VOCs are presented in Sect. 3.1. Results and conclusions drawn from numerical simulations are presented in Sect. 3.2. All errors presented in this work represent $1\sigma$ (66%) and account for the statistical uncertainties calculated by combining the instrument error and the scatter in the relative rate

plots. It is worth noting that the uncertainties quoted on the evaluated literature reference values are often large (up to 35% of the measured value); using these values to place the measured data on an absolute basis therefore limits the final precision of the results in this work. These limitations may not be captured by the quoted uncertainties.

## 3.1 Results from relative rate experiments at 296 K

The measured depletion factor for each VOC in Mixture 1 was plotted against their evaluated rate coefficients using Eq. (1)

(Figure 1). This demonstrates the typical linear relationship between depletion factor and rate coefficient. Mixture 1 contained several aromatic VOCs with multiple methyl substituents, such as the three trimethylbenzene isomers and two isomers of tetramethylbenzene. Multiple, well-evaluated reference reactions (OH + isoprene, OH + pinenes, OH + xylenes) were included as part of this mixture to place the relative rate values on a reliable absolute scale. The rate coefficient results for this mixture are provided in Table 1. All of the measured values were in excellent agreement with the evaluated literature rate coefficients.

In particular, the results for the tetramethylbenzene + OH reactions were in good agreement with their literature counterparts, despite their only being measured once or twice previously (Aschmann et al., 2013; Alarcón et al., 2015).

Figure S2 shows sections of typical ion chromatograms (TIC) obtained for Mixture 2: one chromatogram with the reactor lamp turned off (blue) and one for a sample with the reactor lamp turned on (black). There was a clear, but small, reduction in the peak areas observed with the lamp on relative to the lamp off, corresponding to the depletion in the VOCs due

to reaction with OH. Figure S2 also shows that the observed reduction for the peak assigned to $n$-pentylbenzene was negligible.

Figure 2 shows the relative rate plot for Mixture 2 (see Table 2) with an OH reactivity (in the reactor) of approximately $18\ \mathrm{s}^{-1}$. A linear relationship between depletion factor and $k$ was observed but the scale of the depletion factor was exceptionally low when compared with similar plots for more reactive mixtures (such as Mixture 1 (Fig. 1) or those in Shaw et al., 2018b). Typically, when using synthetic mixtures containing relatively fast reacting isoprenoid species such as monoterpenes, the

measured depletion factors were greater than 0.1 for all VOCs. The depletion factors measured here were equivalent to depletions of less than a few percent; for example, $t$-butyl- and $n$-pentylbenzene recorded depletions of just 0.3% and 0.2% respectively. These small depletions were attributed to the presence of $HO_2$ within the reactor, which represented a greater sink for OH than the VOCs themselves, given their relatively slow reactions with OH.

In an attempt to increase the depletion factor, NO was added to the reactor to convert $HO_2$ to OH (see reaction R4).

There was an observed increase in the measured depletion factors for each of the VOCs corresponding with the increased [NO] in the reactor. The percentage depletions in $t$-butyl- and $n$-pentylbenzene increased to 2.3% and 3.0% respectively when using [NO] = 70 ppb. The addition of NO to the system therefore resulted in increased VOC depletions; this was especially useful





for the slowest reacting VOCs, for which the observed depletions were often below the limit of detection in the absence of NO.

Table 2 provides the estimated rate coefficient values for the reaction of each VOC in Mixture 2 with OH, as calculated using Eq. (1). Measured $k$ values are provided for each of the three experiments using different NO concentrations

together with an overall weighted mean for each of the VOC + OH reactions. Results for each of the ten reactions were in agreement with evaluated literature rate coefficients, within errors. The rate coefficient for $n$-pentylbenzene + OH, $k_{OH}$ (296 K) = 5.3 ($\pm$ 0.7) $\times$ 10$^{-12}$ cm$^3$ molecule$^{-1}$ s$^{-1}$, is one of two determinations reported in this study.

Figure 3 shows the results from a third mixture comprising 10 aromatic VOCs and three simple linear and branched alkanes. The alkanes were used as well-characterised reference reactions to assist with the diagnosis of the depletion factor vs

rate coefficient trend. The final results for this mixture are provided in Table 3. All measured rate coefficients were in agreement with their corresponding evaluated literature values. The measured rate coefficient for the $n$-propylbenzene + OH reaction was larger than the evaluated literature counterpart by approximately 20% but was consistent within errors. This mixture contained a homologous series of mono-substituted aromatic VOC with an increasing alkyl-substituent chain length, from toluene (C$_6$H$_5$CH$_3$) to $n$-pentylbenzene (C$_6$H$_5$(CH$_2$)$_4$CH$_3$). The results for this series of reactions show that the rate

coefficient increased with increasing number of carbon atoms in the aryl group up to four carbon atoms, although the measurement for $k_{OH+n\text{-butylbenzene}}$ had a large derived uncertainty. The rate coefficient for reaction with OH then decreased with the addition of the extra carbon atom from $n$-butylbenzene to $n$-pentylbenzene. The rate coefficient for the $n$-butylbenzene + OH has not been measured previously. The rate coefficient for the $n$-pentylbenzene + OH, $k_{OH}$ (296 K) = 7 ($\pm$ 2) $\times$ 10$^{-12}$ cm$^3$ molecule$^{-1}$ s$^{-1}$, is in agreement with the same measurement in Mixture 2, within 1$\sigma$ uncertainties.

Mixture 3 included the three isomers of diethylbenzene (C$_6$H$_5$(CH$_2$CH$_3$)$_2$), for which no literature rate coefficients for reactions with OH could be found at the time of writing. The averaged measured values, from the five different OH reactivities tested, were ($k_{OH}$ (295 K) in units of 10$^{-12}$ cm$^3$ molecule$^{-1}$ s$^{-1}$): 1,2-diethylbenzene, $k$ = 14 ($\pm$ 4); 1,3-diethylbenzene, $k$ = 22 ($\pm$ 4) and 1,4-diethylbenzene, $k$ = 16 ($\pm$ 4). Values reported here for the diethylbenzene + OH reactions were extrapolated from the depletion factor vs. $k$ relationship; their measured depletions were all greater than the depletion in the fastest reacting

reference compound. There may therefore be a larger extent of uncertainty in their values than that calculated, as the $OH_{exp}$ trend beyond $o$-xylene was unmeasured.

A fourth synthetic gas mixture, containing aliphatic VOCs with a range of sizes, branching and OH reaction rate coefficients, was also measured using the multivariate relative rate method. Figure 4 shows the relative rate plot for this VOC mixture. A clear linear relationship was observed between depletion factor and literature rate coefficient. Rate coefficients for

the reactions of OH with three VOCs (2-methylheptane, 2-methylnonane and ethylcyclohexane), were measured for the first time. Table 4 provides a list of compounds in this mixture, along with their measured rate coefficient and evaluated literature reference rate coefficient, where available.



## 3.2 Comparisons to structure activity relationships

Structure activity relationships (SARs) provide a method for estimating rate coefficients and branching ratios for the reactions of compounds based on their structural properties. SAR derived parameters, when based on large subsets of the kinetic catalogue, may be used as comparisons for the measured values in this work in the absence of, or in parallel with, literature

experimental results (Vereecken et al., 2018). Recently, Jenkin et al. (2018a) developed an updated SAR for estimating rate coefficients for reactions of aromatic organic compounds with OH. The SAR was based on a preferred  data set of kinetic measurements for twenty-five monocyclic and forty-two oxygenated aromatic hydrocarbons, and returned results with no significant bias, and an average uncertainty of 15% relative to the observational dataset; hence providing a reasonably accurate representation of aromatic reactivity with OH. A similarly constructed SAR, for the estimation of rate coefficients for reactions

involving aliphatic compounds, can be found in Jenkin et al. (2018b).

Figure 5 provides a comparison between the measured rate coefficients from this work and the calculated rate coefficients from Jenkin et al. (2018a) (for the reactions of aromatic VOCs with OH), and Jenkin et al. (2018b) (for the reactions of aliphatic VOCs with OH). All of the values measured using the multivariate relative rate technique were in good agreement with those estimated by the SARs. This provides additional validation of the SARs proposed in Jenkin et al. (2018a and 2018b).

The five reactions involving aromatic compounds for which no literature measured values existed at the time of writing, were also in good agreement with those estimated using the SAR. Table S1 provides additional SAR derived parameters for these five aromatic reactions, and the three newly measured aliphatic reactions, using alternative SARs described by Kwok and Atkinson (1995) and Ziemann and Atkinson (2012). These SARs performed relatively poorly when simulating the aromatic + OH reaction rate coefficients but performed well for the aliphatic + OH reaction rate coefficients.

Figure 6 shows a more detailed analysis of the measured $k_{OH}$ (295 K) values for the reactions between OH and a homologous series of $n$-alkylbenzenes. Measurements from this work, and a previous study utilising the multivariate relative rate technique, are shown alongside the trend predicted by the Jenkin et al. (2018a) SAR. The SAR generally predicts a small increase in rate coefficient with increasing chain length. Many of the experimental results were in agreement with the SAR when accounting for uncertainties.

Figure 7 shows a similar plot for comparison of the measured rate coefficients for the diethylbenzene isomers with those predicted by the Jenkin et al. (2018a) SAR. This plot further illustrates the good agreement between experimental results from this work and the rate coefficients estimated using the SAR. The experimental results were marginally greater than the predicted values but were in agreement when accounting for uncertainties, despite the lack of previous literature measurements for the diethylbenzene compounds. The proposed aromatic OH SAR presented in Jenkin et al. (2018a) therefore agrees well

with the measurements for all aromatic compounds measured as part of this work.



### 3.2 Interpretations from numerical simulations

Numerical simulations were conducted to aid in the evaluation of the experimental results obtained as part of this work for the aromatic systems. Simulations were performed using the Kintecus software package (Ianni, 2017).

The first set of simulations aimed to assess the significance of the thermal decomposition of the OH-aromatic adducts (as depicted in Figure S5) at different temperatures (Calvert et al., 2002). The chemical model for these simulations used Arrhenius expressions for the initial reactions of OH with benzene, toluene, and *m*- and *p*-xylene (Atkinson and Arey, 2003; Atkinson et al., 2006). The chemical scheme included two subsequent pathways; one in which the OH-aromatic adducts decayed back into the initial aromatics (and OH), and one in which they reacted with $O_2$. Arrhenius expressions for each of these reactions were included in the model (Perry et al., 1977; Knispel et al., 1990; Koch et al., 2007). Minor H-atom abstraction pathways were neglected. The initial OH concentration was kept constant at $2 \times 10^{11}$ molecules $cm^{-3}$. The concentration of $O_2$ within the experimental reactor was not measured experimentally so an estimate of $1 \times 10^{17}$ molecules $cm^{-3}$ of $O_2$ (equivalent to an $N_2$ purity of approximately 99%) was used during the simulations.

Allowing for the decay of the OH-aromatic adduct (at $T = 300$ K) resulted in the correctly simulated relationship between depletion factor and literature $k_{OH}$ value for all four aromatic species, with an $R^2$ value of 0.9995. Figure 8 shows that this relationship was sensitive to the concentration of $O_2$ in the simulation, with the $R^2$ value decreasing to 0.856 at $[O_2] = 1 \times 10^{15}$ molecules $cm^{-3}$. Therefore, at the experimental temperature ($T = 296$ K), significant back-decomposition of the OH-aromatic adduct should have been avoided, given the assumed 1% $O_2$ content ($10^{17}$ molecules $cm^{-3}$) within the reactor.

At elevated temperatures (e.g. $T = 350$ K), the decay of the aromatic adducts proceeds at a quicker rate. The relationship between depletion factor and literature $k$ value was poorly simulated, even when using the assumed concentration of $O_2$. The simulated depletion factors for some of the aromatic species were also well below the limit of detection for experimental observations, owing to the rapid reformation of the original aromatic species. This suggests that there is a threshold temperature above which significant thermal decomposition of the OH-aromatic adducts would occur, limiting the potential for experimental results.

The thermal instability of the primary product of the aromatic + OH reactions therefore complicated the relative rate nature of these experiments. The potential reformation of the original aromatic species meant that their measured concentrations after reaction with OH may not have been dependent on just their rate coefficient for reaction with OH. However, the numerical simulations performed as part of this work yielded an important conclusion: at the reactor temperatures used during experimentation, the extent of OH-aromatic decay was largely insignificant as long as approximately $10^{17}$ molecules $cm^{-3}$ of $O_2$ were present in the reactor, which is assumed to be the case.

A second set of simulations were performed to model the impact that NO had on the chemical mechanisms occurring within the reactor. The chemical scheme for these simulations incorporated the oxidation of the aromatic compounds toluene, ethylbenzene, *n*-propylbenzene, isopropylbenzene, *m*- and *o*-xylene and 2-, 3- and 4-ethyltoluene, as adapted from the Master Chemical Mechanism (Jenkin et al., 2003; Bloss et al., 2005). All reactions involving $O_3$ and $NO_3$ were removed from the





chemical model as these oxidants were assumed to be absent from the experimental reactor. Reactions involving $RO_2$ were also removed from the chemical model for simplification reasons. Basic $HO_x/NO_x$ chemistry was included to yield a simplified chemical scheme comprising a total of 160 reactions and 88 different species.

Reactions involving primary OH (henceforth referred to as *pOH*) were separated from reactions involving secondary OH (*sOH*) within the model. *pOH* corresponds to the OH generated as a direct result of $H_2O$ photolysis. *sOH* refers to the OH generated as a result of the reaction between NO and $HO_2$ (R4).

Figure S6 shows that adding NO to the reactor increased the simulated extent of aromatic VOC depleted through reactions with OH. A total of 6.1% of the total aromatic VOC reacted with OH in the case without NO compared to 8.2% in simulations with 30 ppb of NO. This correlated well with evidence from experiments in which the depletion factors for individual VOCs increased after the addition of NO to the reactor. However, the simulated proportion of reacted VOC decreased after addition of 70 ppb of NO. This was not observed experimentally. This may suggest that there is an optimum amount of NO to include in the reactor, above which the sinks of OH to reactions with NO and $NO_2$ begins to dominate. This critical point was not surpassed experimentally, hence leading to the observed trend where increased [NO] led to an increase in depletion factor.



## 4 Atmospheric implications and conclusions

The multivariate relative rate technique was successfully adapted to measure the relative reaction rates of twenty different mono-aromatic VOC with OH (at 296 K). Fifteen of these reactions have had rate coefficients reported at least once previously, with the measurements reported here in good agreement with this existing limited experimental dataset. Five of the measured

rate coefficients are reported for the first time. These are (in units of $10^{-12}$ cm$^3$ molecule$^{-1}$ s$^{-1}$): *n*-butylbenzene, 11 (± 4); *n*-pentylbenzene, 7 (± 2); 1,2-diethylbenzene, 14 (± 4); 1,3-diethylbenzene, 22 (± 4) and 1,4-diethylbenzene, 16 (± 4). The OH rate coefficients for 12 straight and branched chain aliphatic species were also measured using the multivariate relative rate technique, three of which are not available in the literature. The measured OH reaction rate coefficients for these were (in units of $10^{-12}$ cm$^3$ molecule$^{-1}$ s$^{-1}$): 2-methylheptane, 9.1 (± 0.3); 2-methylnonane, 11.0 (± 0.3) and ethylcyclohexane, 14.4 (± 0.3).

The 296 K measurements were in agreement with the available, but often limited, experimental dataset.

Measurements of slower reactions (relative to many of those measured in Shaw et al., 2018b) proved difficult owing to the limited depletions observed for some of the VOCs. A simple chemical modification to the basic setup successfully increased the extent of the measured depletions allowing for the evaluation of the rates of reaction for slower reacting compounds. The addition of NO to flow reactors is therefore advised for any future work performing this type of self-

consistent, multivariate relative rate measurement, particularly when the rates of reaction for the targeted reactive species are below $2.0 \times 10^{-11}$ cm$^3$ molecule$^{-1}$ s$^{-1}$. Model sensitivity simulations on the thermal stability of the OH-aromatic adduct intermediate species suggested that there should be limited decomposition to reform the original aromatic precursors under the applied experimental conditions.

Atmospheric lifetimes for the eight VOCs for which rate coefficients for reaction with OH were measured for the first

time, were calculated using a 24-hour average atmospheric concentration of [OH] = $1 \times 10^6$ molecules cm$^{-3}$. Estimated lifetimes due to reaction with OH ($\tau_{OH}$) for the five aromatic compounds were: *n*-butylbenzene, 25 hours; *n*-pentylbenzene, 40 hours; 1,2-diethylbenzene, 20 hours; 1,3-diethylbenzene, 13 hours and 1,4-diethylbenzene, 17 hours. Given that these aromatic compounds are relatively large and react rapidly with OH (when compared with toluene), they would be expected to have reasonably high photochemical ozone creation potentials (POCPs). Estimated lifetimes due to reaction with OH for 2-

methylheptane, 2-methylnonane and ethylcyclohexane were 31, 25 and 19 hours respectively. It should be noted that the lifetimes indicated here have limited applicability in atmospheric environments with substantially different OH concentrations, and that they may be as short as a few hours in environments with particularly high concentrations of OH. Comprehensive chemical modelling would need to be undertaken to fully understand the behaviour of these compounds in different environments, but this is beyond the scope of this work.

The majority of the rate coefficients measured as part of this work were in good agreement with the recently developed SARs by Jenkin et al. for estimating aromatic and aliphatic OH kinetics (Jenkin et al., 2018a; Jenkin et al., 2018b) validating the use of theoretical calculation of OH rate coefficients for reactions involving these compounds. This work highlights that more experimental and theoretical work is required to understand the OH oxidation kinetics and mechanisms of high alkyl





substituted aromatic hydrocarbons in order to extend the kinetic database for more accurate SAR parameter estimations and to assess their impact on atmospheric composition.



**Data availability**

Raw data is available upon request.

**Author contribution**

ARR and TJD planned the overall project; JTS and TJD designed experiments; JTS conducted all experiments and performed

5   the analysis; JTS and TJD prepared the manuscript with contributions from all authors.

**Competing interests**

The authors declare that they have no conflict of interest.

**Acknowledgements**

The authors would like to thank Martyn Ward, Chris Mortimer and Chris Rhodes for their assistance and technical support

10   throughout the project. We are also grateful to the Natural Science Research Council (NERC) for funding this work under

grant numbers NE/M013448/1, NE/J008990/1, and NE/J008532/1. We would also like to thank NERC for the provision and

funding of a PhD for JTS as part of the SPHERES DTP scheme.



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



**Tables**

**Table 1: Results of relative rate experiments, with evaluated literature data, for each VOC in Mixture 1, ordered by evaluated literature $k$ value. The measured values presented here are the weighted averages of five individual experiments conducted with different mixture OH reactivities.**

| Name | Measured $k$ (296 K) / $10^{-12}$ cm$^3$ molecule$^{-1}$ s$^{-1}$ | Evaluated literature $k$ (298 K) / $10^{-12}$ cm$^3$ molecule$^{-1}$ s$^{-1}$ | Reference | Number of literature measurements |
|---|---|---|---|---|
| Isoprene | $102 \pm 4$ | $100 \ (^{+15}_{-13})$ | Atkinson et al., 2006 | 25+ |
| β-pinene | $74 \pm 8$ | $79 \pm 20$ | Atkinson and Arey, 2003 | 10 |
| 1,2,3,5-tetramethylbenzene | $62 \pm 9$ | $62.4 \pm 0.8$ | Alarcón et al., 2015 | 1 |
| 1,3,5-trimethylbenzene | $60 \pm 5$ | $57 \pm 11$ | Calvert et al., 2002 | 5 |
| 1,2,4,5-tetramethylbenzene | $59 \pm 12$ | $55.5 \pm 3.4$ | Aschmann et al., 2013 | 2 |
| α-pinene | $53 \pm 8$ | $53 \ (^{+22}_{-15})$ | Atkinson et al., 2006 | 9 |
| 1,2,4-trimethylbenzene | $34 \pm 3$ | $33 \pm 8$ | Calvert et al., 2002 | 5 |
| 1,2,3-trimethylbenzene | $38 \pm 4$ | $33 \pm 8$ | Calvert et al., 2002 | 5 |
| $m$-xylene | $21 \pm 3$ | $23 \pm 3$ | Calvert et al., 2002 | 15 |
| $o$-xylene | $10 \pm 4$ | $13 \pm 3$ | Calvert et al., 2002 | 10 |





**Table 2: Results of relative rate experiments, with evaluated literature data, for each VOC in Mixture 2, ordered by evaluated literature *k* value. The results of experiments using different concentrations of NO are presented, along with a mean measured k value for each VOC, weighted to the errors in the individual NO experiments. Each iteration with a different [NO] comprised three individual experiments with different OH reactivity of mixture injected into the reactor. Figure S3 shows the relative rate plots for Mixture 2 with an OH reactivity (in the reactor) of 18 s⁻¹ and NO concentrations of 0, 30 and 70 ppb.**

| Name | Measured $k$ (296 K) / $10^{-12}$ cm³ molecule⁻¹ s⁻¹ | | | | Evaluated literature $k$ (298 K) / $10^{-12}$ cm³ molecule⁻¹ s⁻¹ | Reference | Number of literature measurements |
|---|---|---|---|---|---|---|---|
| | [NO] = 0 ppb | [NO] = 30 ppb | [NO] = 70 ppb | Weighted average | | | |
| *m*-xylene | 20.6 ± 0.7 | 19.6 ± 0.4 | 19.8 ± 1.3 | 19.9 ± 0.5 | 23 ± 3 | Calvert et al. 2002 | 15 |
| 3-ethyltoluene | 19.7 ± 0.9 | 21.7 ± 0.4 | 19.4 ± 1.0 | 21.1 ± 1.2 | 19 ± 7 | Calvert et al. 2002 | 2 |
| *o*-xylene | 12.3 ± 1.3 | 12.4 ± 0.4 | 12.3 ± 0.6 | 12.4 ± 0.1 | 13 ± 3 | Calvert et al. 2002 | 10 |
| 4-ethyltoluene | 14 ± 2 | 14.8 ± 0.5 | 14.1 ± 1.2 | 14.6 ± 0.3 | 12 ± 4 | Calvert et al. 2002 | 2 |
| 2-ethyltoluene | 13.4 ± 1.0 | 15.0 ± 1.1 | 12.7 ± 0.1 | 12.7 ± 0.2 | 12 ± 4 | Calvert et al. 2002 | 2 |
| Ethylbenzene | 5 ± 2 | 6.0 ± 0.3 | 6.7 ± 0.5 | 6.1 ± 0.4 | 7.0 ± 2 | Calvert et al. 2002 | 3 |
| Isopropylbenzene | 6.5 ± 0.5 | 6.4 ± 0.4 | 6.5 ± 0.6 | 6.45 ± 0.02 | 6.3 ± 2 | Calvert et al. 2002 | 3 |
| *n*-propylbenzene | 8.9 ± 0.7 | 8.7 ± 0.6 | 7 ± 2 | 8.7 ± 0.5 | 5.8 ± 1.5 | Calvert et al. 2002 | 3 |
| Toluene | 5 ± 2 | 0 ± 2 | 7 ± 2 | 4 ± 4 | 5.6 (±1.5) | Atkinson et al. 2006 | 18 |
| *t*-butylbenzene | 3.5 ± 1.1 | 3.4 ± 0.5 | 3.1 ± 0.7 | 3.3 ± 0.1 | 4.5 ± 2 | Calvert et al. 2002 | 2 |
| *n*-pentylbenzene | 3 ± 5 | 5.4 ± 0.4 | 1 ± 4 | 5.3 ± 0.7 | | | |



**Table 3: Results of relative rate experiments, with evaluated literature data, for each VOC in Mixture 3, ordered by evaluated literature $k$ value. The measured values presented here are the weighted averages of five individual experiments conducted with different mixture OH reactivities and 40 ppbv NO. Five VOCs had no reported data for their reactions with OH.**

| Name | Measured $k$ (296 K) / $10^{-12}$ cm$^3$ molecule$^{-1}$ s$^{-1}$ | Evaluated literature $k$ (298 K) / $10^{-12}$ cm- molecule$^{-1}$ s$^{-1}$ | Reference | Number of literature measurements |
|---|---|---|---|---|
| *o*-xylene | $13.0 \pm 1.2$ | $13 \pm 3$ | Calvert et al. 2002 | 10 |
| 2-ethyltoluene | $12.2 \pm 0.8$ | $12 \pm 4$ | Calvert et al. 2002 | 2 |
| *n*-decane | $10.7 \pm 0.7$ | $11 \pm 2$ | Atkinson, 2003 | 6 |
| *n*-nonane | $10.1 \pm 1.2$ | $10 \pm 2$ | Atkinson, 2003 | 9 |
| Ethylbenzene | $6.6 \pm 0.5$ | $7 \pm 2$ | Calvert et al. 2002 | 3 |
| *n*-propylbenzene | $6.8 \pm 0.3$ | $5.8 \pm 1.5$ | Atkinson, 2003 | 3 |
| Toluene | $4 \pm 2$ | $5.6 (^{+1.5}_{-1.2})$ | Atkinson et al., 2006 | 18 |
| 2,2,3-trimethylbutane | $6 \pm 6$ | $3.8 \pm 1.0$ | Atkinson, 2003 | 6 |
| 1,3-diethylbenzene | $22 \pm 4$ | | | |
| 1,4-diethylbenzene | $16 \pm 4$ | | | |
| 1,2-diethylbenzene | $14 \pm 4$ | | | |
| *n*-butylbenzene | $11 \pm 4$ | | | |
| *n*-pentylbenzene | $7 \pm 2$ | | | |



**Table 4: Results of relative rate experiments, with evaluated literature data, for each VOC in Mixture 4, ordered by evaluated literature $k$ value.**

| Name | Measured $k$ (295 K) / $10^{-12}$ cm$^3$ molecule$^{-1}$ s$^{-1}$ | Evaluated literature $k$ (298 K) / $10^{-12}$ cm$^3$ molecule$^{-1}$ s$^{-1}$ | Reference | Number of literature measurements |
|---|---|---|---|---|
| Cyclooctane | $13.7 \pm 0.3$ | $13 \pm 7$ | Atkinson, 2003 | 2 |
| $n$-undecane | $12.3 \pm 0.3$ | $12 \pm 2$ | Atkinson, 2003 | 2 |
| Cycloheptane | $11.4 \pm 0.3$ | $12 \pm 3$ | Atkinson, 2003 | 3 |
| $n$-decane | $10.3 \pm 0.3$ | $11 \pm 2$ | Atkinson, 2003 | 6 |
| $n$-nonane | $11.0 \pm 0.3$ | $10 \pm 2$ | Atkinson, 2003 | 9 |
| $n$-octane | $8.8 \pm 0.3$ | $8 \pm 2$ | Atkinson, 2003 | 6 |
| 2-methylpentane | $4.6 \pm 0.3$ | $5.2 \pm 1.3$ | Atkinson, 2003 | 4 |
| 3-methylpentane | $5.3 \pm 0.3$ | $5.2 \pm 1.3$ | Atkinson, 2003 | 3 |
| 2,2,3-trimethylbutane | $3.9 \pm 0.3$ | $3.8 \pm 1.0$ | Atkinson, 2003 | 6 |
| 2-methylheptane | $9.1 \pm 0.3$ | | | 0* |
| 2-methylnonane | $11.0 \pm 0.3$ | | | 0 |
| Ethylcyclohexane | $14.4 \pm 0.3$ | | | 0* |

\* Shaw et al. (2018) reported $k_{OH}$(323 K) for these compounds and no literature $k_{OH}$ value was found at 298 K.





**Figures**

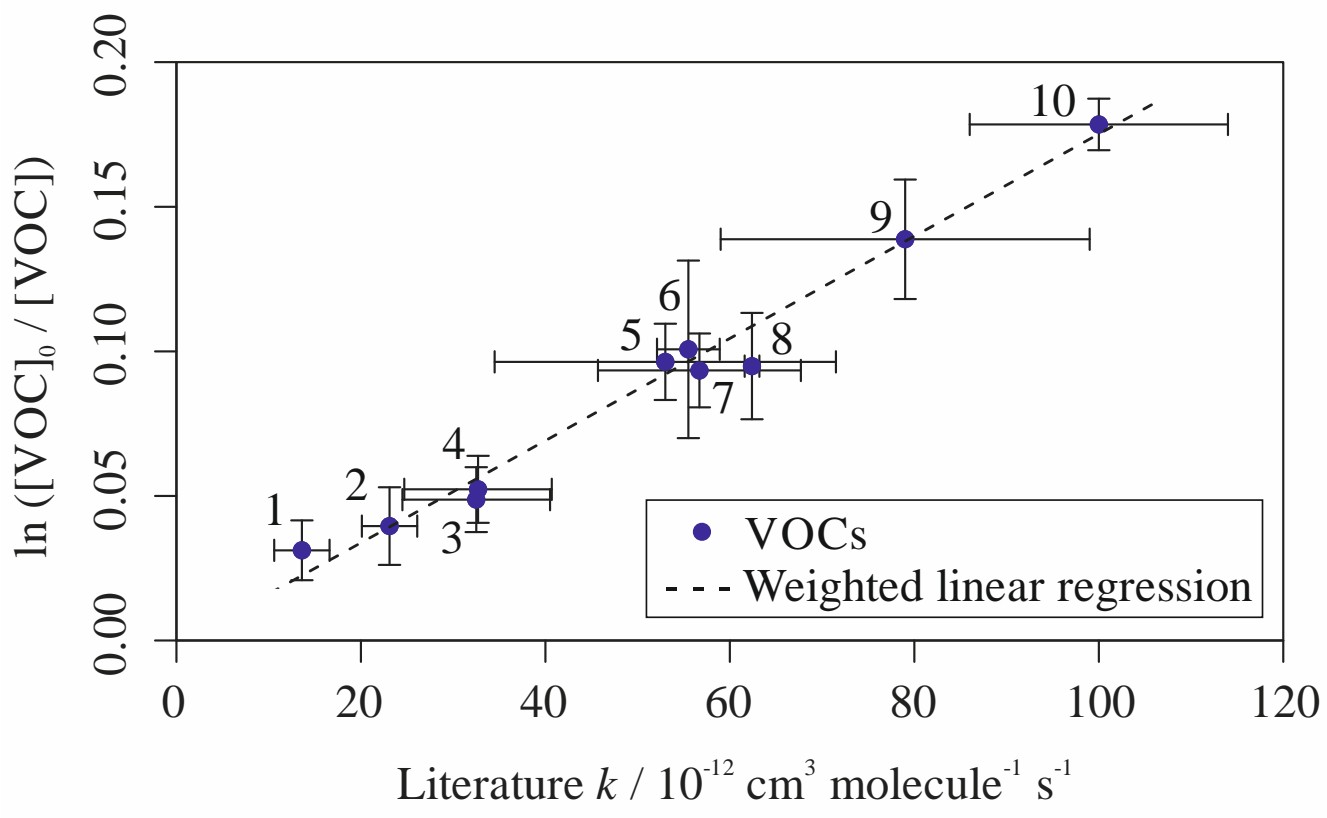

**Figure 1: Relative rate plot for Mixture 1 (OH reactivity of 48 s$^{-1}$) in the absence of NO at 296 ($\pm$ 2) K. Compounds with a reference rate coefficient for reaction with OH using evaluated literature values. Error bars on the *y*-axis, equal to 1 standard error, were calculated by combining the standard error in peak areas for eight lamp-off and eight lamp-on samples. Error bars on the *x*-axis were typically large (approximately $\pm$ 20-30 %) and accounted for deviations from the predicted trend for all VOCs. A weighted (to the uncertainty in the y values) linear fit was used to generate the slope with *OHexp* = 1.8 ($\pm$ 0.1) $\times$ 10$^9$ molecules cm$^{-3}$ s and R$^2$ = 0.980. The VOCs can be identified as follows: 1, *o*-xylene; 2, *m*-xylene; 3, 1,2,4-trimethylbenzene; 4, 1,2,3-trimethylbenzene; 5, α-pinene; 6, 1,2,4,5-tetramethylbenzene; 7, 1,3,5-trimethylbenzene; 8, 1,2,3,5-tetramethylbenzene; 9, β-pinene; 10, isoprene.**



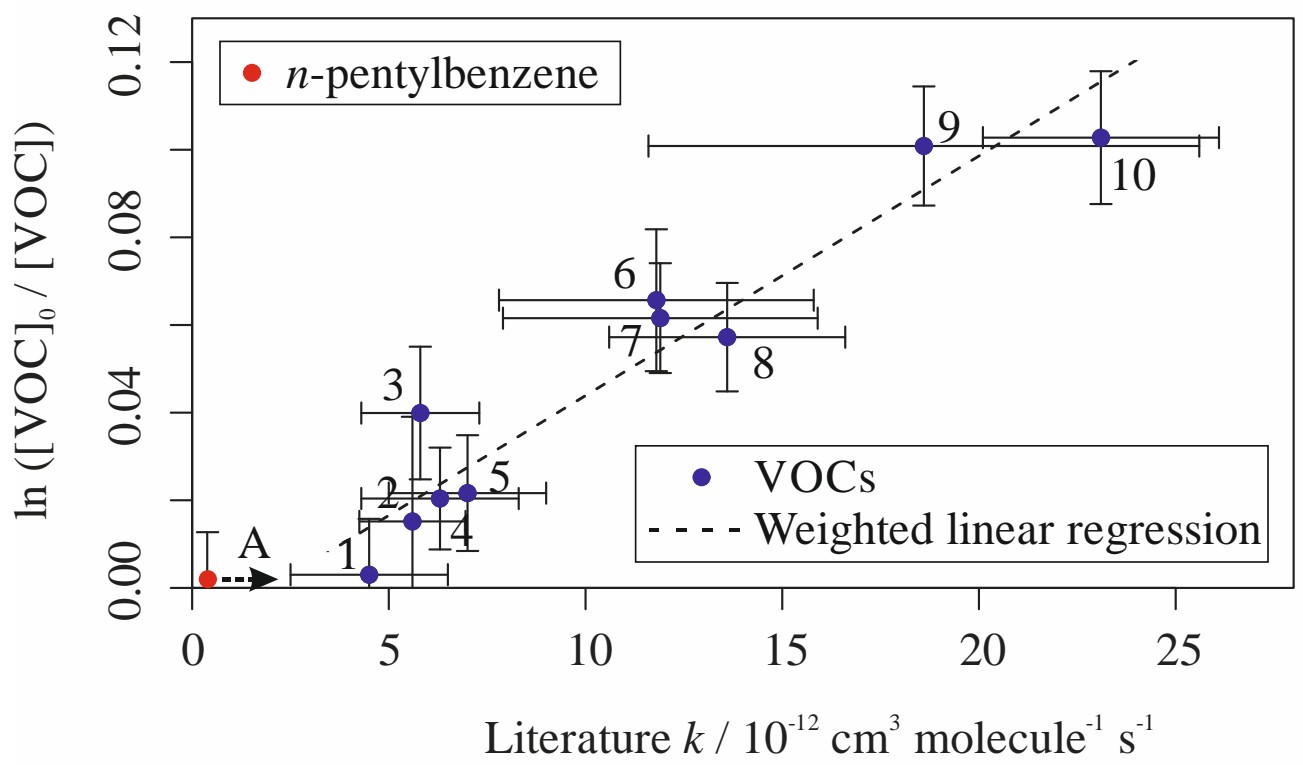

**Figure 2: Relative rate plot for Mixture 2 (OH reactivity of 18 s⁻¹) in the absence of NO at 296 (± 2) K. Compounds with a reference rate coefficient for reaction with OH were plotted using evaluated literature values. Error bars on the *y*-axis, equal to 1 standard error, were calculated by combining the standard error in peak areas for six lamp-off and six lamp-on samples. Error bars on the *x*-axis were typically large (approximately ± 20-30 %) and accounted for deviations from the predicted trend for most VOCs. A weighted (to the uncertainty in the y values) linear fit was used to generate the slope with *OHexp* = 5.5 (± 0.6) × 10⁹ molecules cm⁻³ s and R² = 0.899. Data for *n*-pentylbenzene (A) was not used in the calculation of *OHexp*. The VOCs can be identified as follows: 1, *t*-butylbenzene; 2, toluene; 3, *n*-propylbenzene; 4, isopropylbenzene; 5, ethylbenzene; 6, 2-ethyltoluene; 7, 4-ethyltoluene; 8, *o*-xylene; 9, 3-ethyltoluene; 10, *m*-xylene. The dashed arrow indicates that the *k* value for the target compound is determined by where it would fall on the measured *OHexp* curve.**





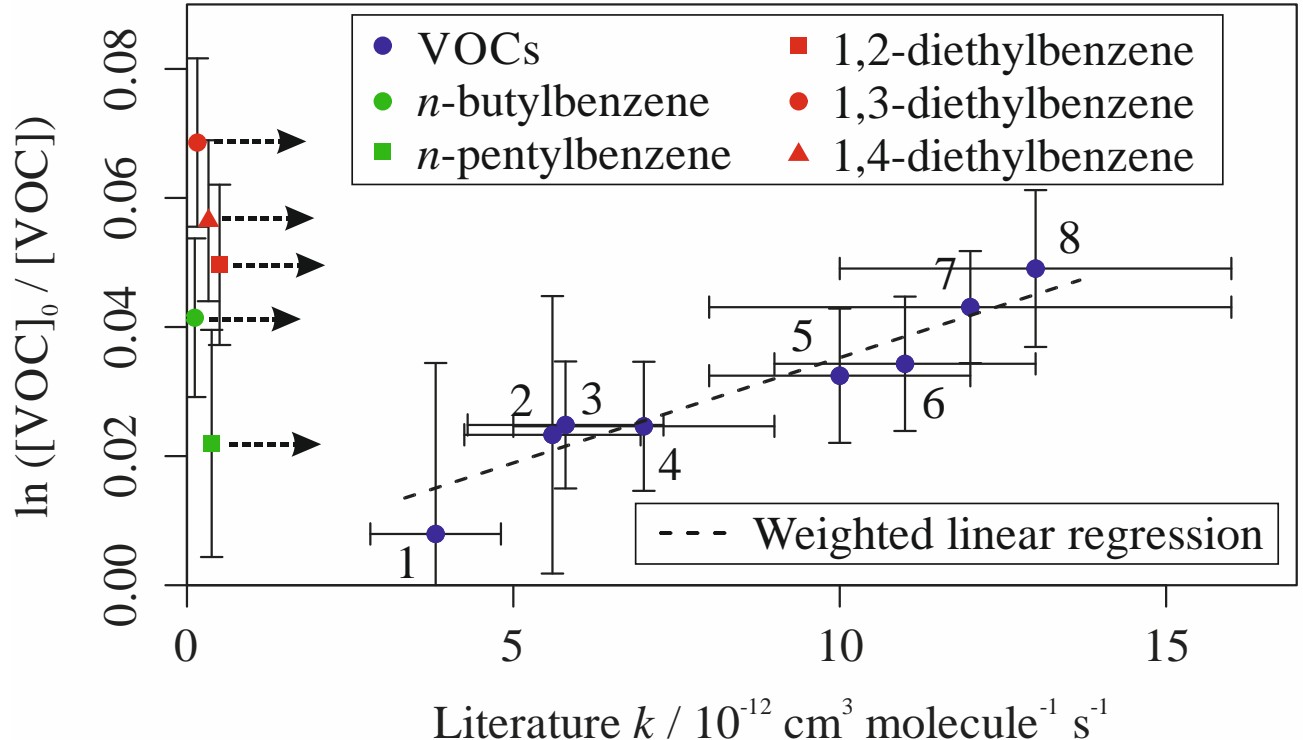

**Figure 3: Relative rate plot for Mixture 3 (OH reactivity of 6 s$^{-1}$) and 40 ppb NO at 296 (± 2) K. Compounds with a reference rate coefficient for reaction with OH were plotted using evaluated literature values. Error bars on the $y$-axis, equal to 1 standard error, were calculated by combining the standard error in peak areas for eight lamp-off and eight lamp-on samples. Error bars on the $x$-axis were typically large (approximately ± 20-30 %) and accounted for deviations from the predicted trend for most VOCs. A weighted (to the uncertainty in the y values) linear fit was used to generate the slope with $OHexp$ = 3.3 (± 0.4) × 10$^9$ molecules cm$^{-3}$ s and R$^2$ = 0.890. Data for 1,2-, 1,3- and 1,4-diethylbenzene and $n$-butyl and $n$-pentylbenzene were not used in the calculation of $OHexp$. The VOCs can be identified as follows: 1, 2,2,2-trimethylbutane; 2, toluene; 3, $n$-propylbenzene; 4, ethylbenzene; 5, $n$-nonane; 6, $n$-decane; 7, 2-ethyltoluene; 8, $o$-xylene. The dashed arrows indicate that the $k$ values for the target compounds are determined by where they would fall on the measured $OHexp$ curve.**



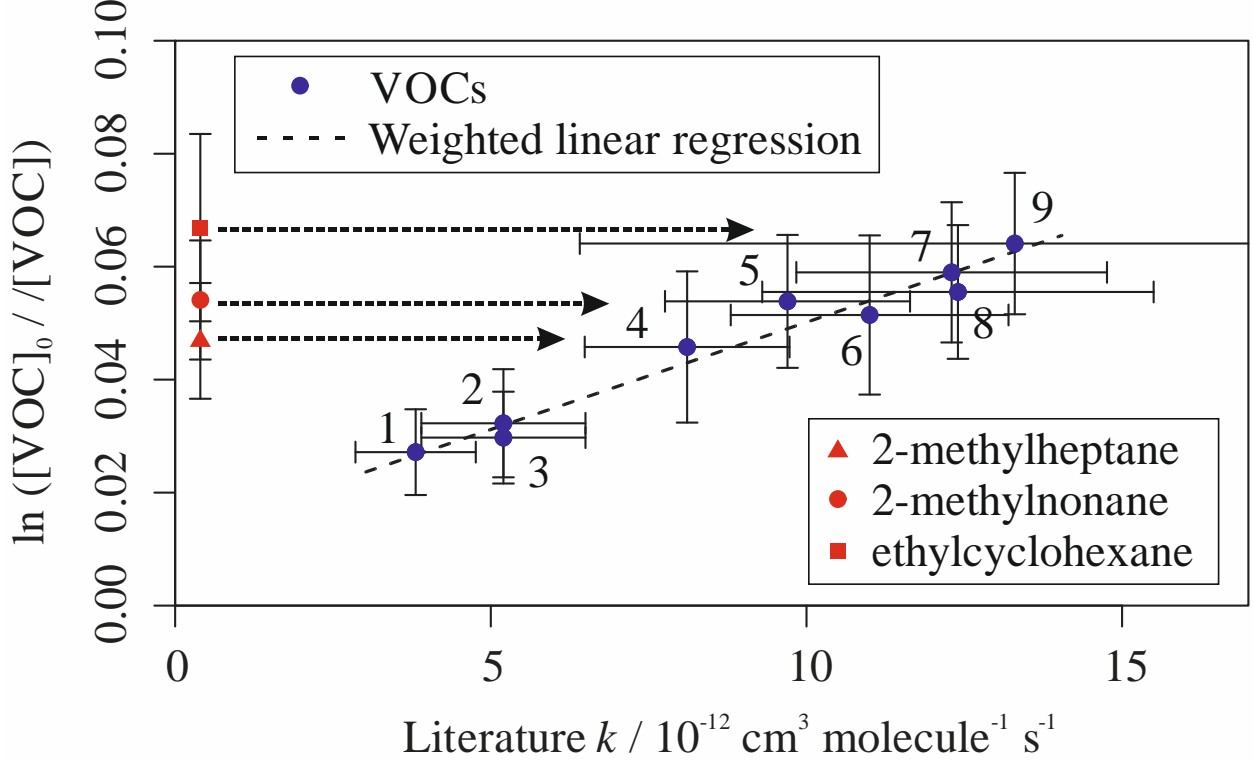

**Figure 4: Relative rate plot for Mixture 4 (OH reactivity of 43 s⁻¹) and 50 ppb NO, at 296 K. Compounds with a reference rate coefficient for reaction with OH were plotted using their evaluated literature values. Errors on the *y*-axis, equal to one standard error, were calculated by combining the standard error in peak areas for six lamp-off and six lamp-on samples. Error bars on the *x*-axis were typically large and accounted for deviations from the line for all VOCs. A weighted (to the uncertainty in the *y*-values) linear fit was used to generate the slope, with a value of $OH_{exp}$ = 3.8 (± 0.2) × 10⁹ molecules cm⁻³ s and R² = 0.971. Data for 2-methylheptane, 2-methylnonane and ethylcyclohexane, which had no literature *k* (298 K) values, were not used in the calculation of the fit. The VOCs can be identified as follows; 1) 2,2,3-trimethylbutane; 2) 2-methylpentane; 3) 3-methylpentane; 4) *n*-octane; 5) *n*-nonane; 6) *n*-decane; 7) *n*-undecane; 8) cycloheptane; 9) cyclooctane. The dashed arrows indicate that the *k* values for the target compounds are determined by where they would fall on the measured $OH_{exp}$ curve.**

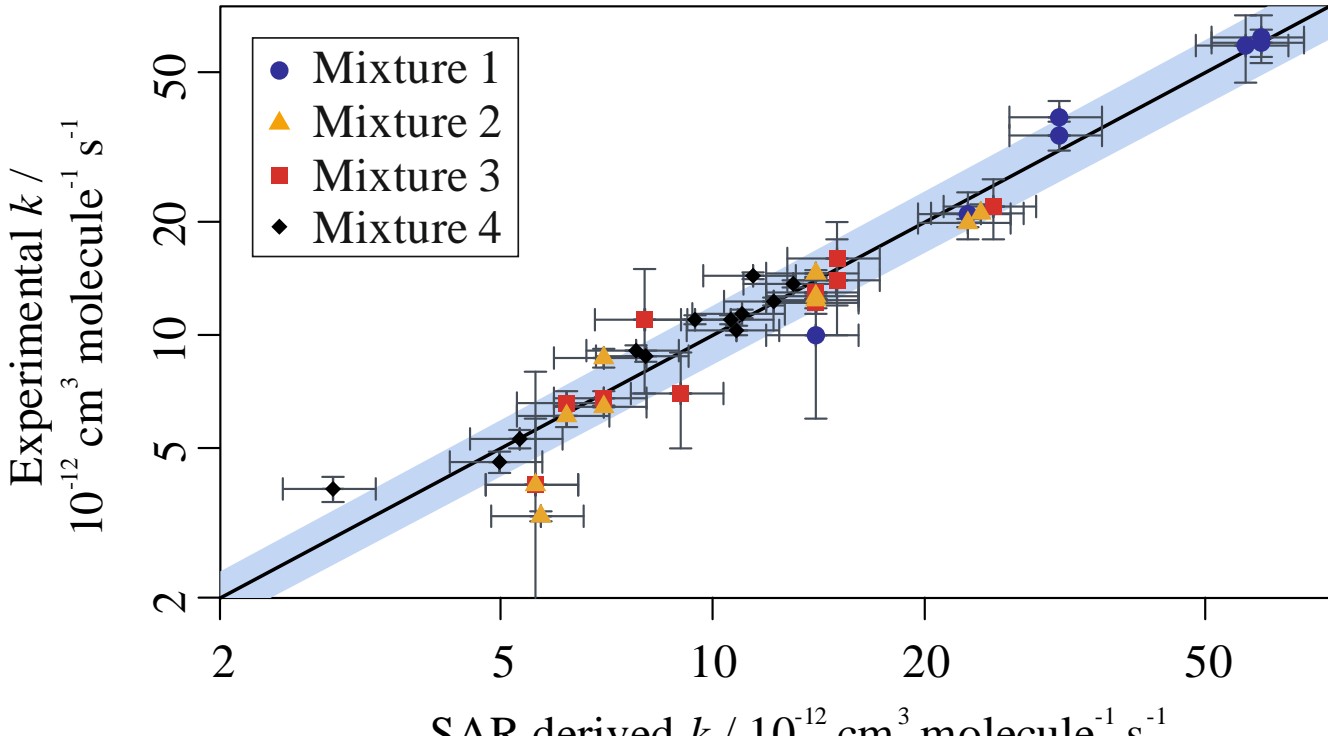

**Figure 5: Measured VOC + OH rate coefficients plotted against SAR derived rate coefficients for all compounds in Mixtures 1, 2, 3 and 4. The grey shaded area demonstrates a 25% uncertainty in the 1:1 black gradient line. Most data fall well within this bound. The structure activity relationships used can be found in Jenkin et al. 2018a and Jenkin et al., 2018b. Outliers are; 2,2,3-trimethylbenzene, toluene, *t*-butylbenzene, *n*-butylbenzene, *n*-pentylbenzene and *o*-xylene. Most outliers have large derived uncertainties in their measured value.**





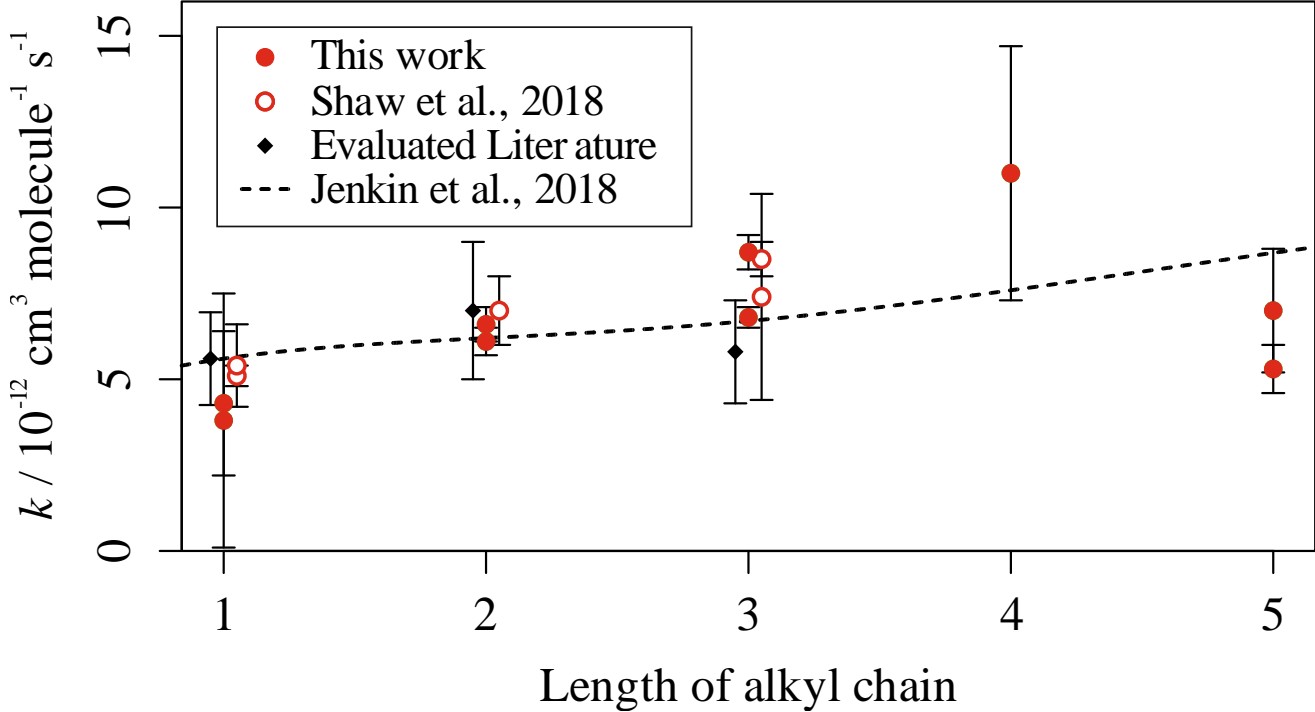

**Figure 6. Experimentally derived (data points; see Table 2 and 3) and estimated SAR-derived (line; see Jenkin et al. (2018a)) $k$ values for the reactions between OH and a homologous series of $n$-alkylbenzenes with different numbers of carbon atoms in the alkyl chain. Data from this, and other multivariate relative rate studies, are shown in red.**





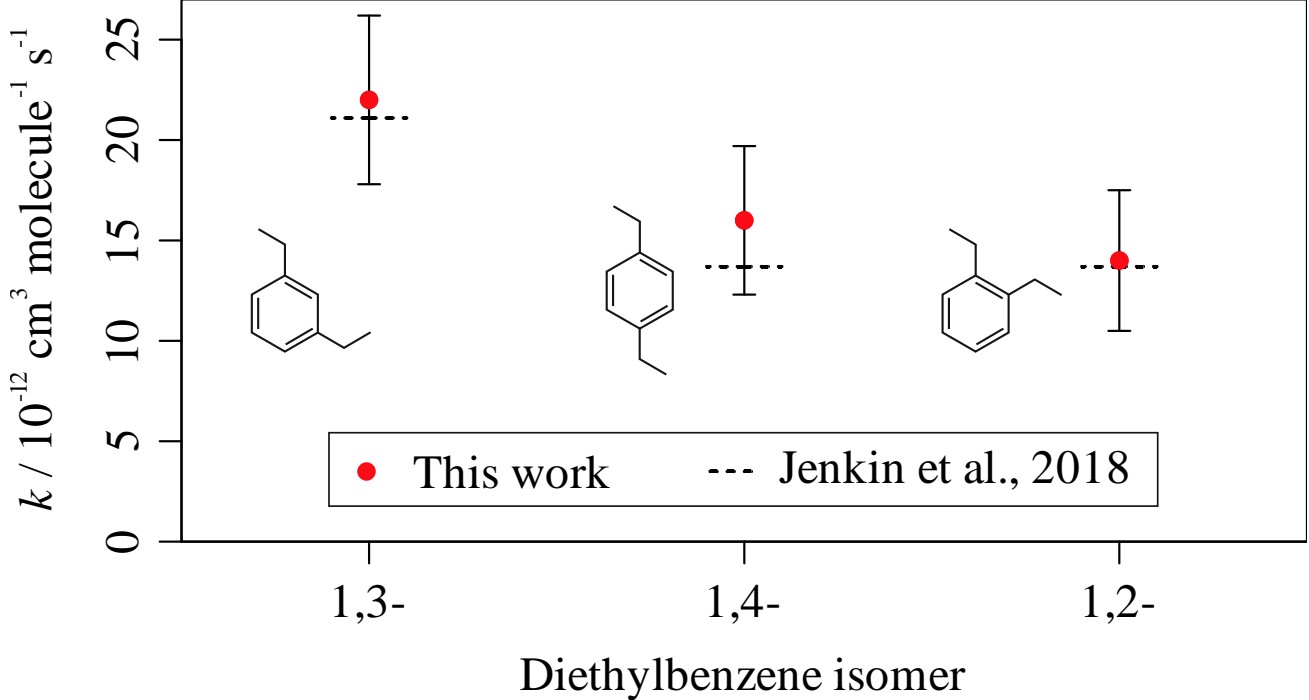

**Figure 7. Experimentally derived (data points) and estimated SAR-derived (lines) *k* values for the reactions between OH and the three diethylbenzene isomers. Data from this work are shown in red.**

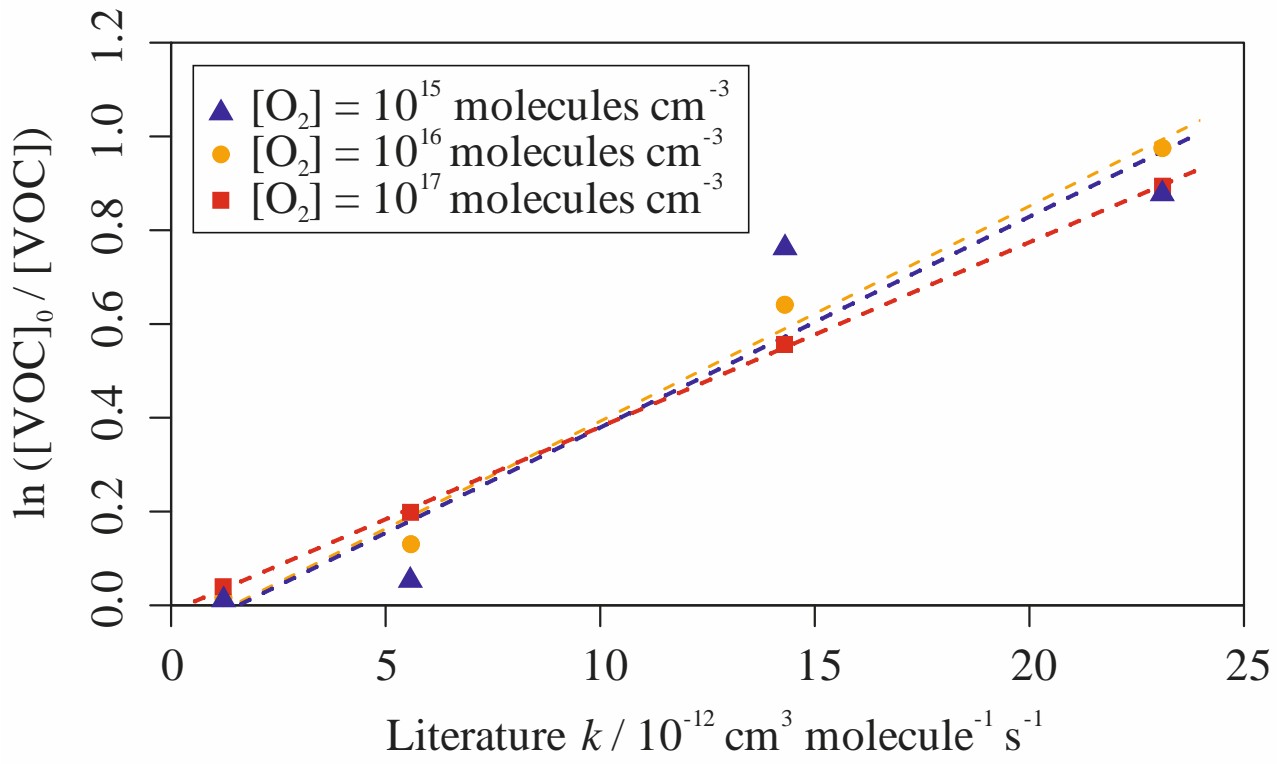

**Figure 8. Simulated relative rate plot for four aromatic VOCs when simulating different concentrations of $O_2$ in the reactor at 300 K. From left to right, the data points can be identified as; benzene, toluene, *p*-xylene and *m*-xylene. The $R^2$ values were 0.856, 0.981 and 1.00 for $[O_2] = 10^{15}$, $10^{16}$ and $10^{17}$ molecules cm$^{-3}$ respectively. The relationship between depletion factor and *k* value improved with increasing $[O_2]$ suggesting that greater concentrations would be more likely to provide more accurate results.**