# Peer review of "Rate coefficients for reactions of OH with aromatic and aliphatic volatile organic compounds determined by the Multivariate Relative Rate Technique"

_Atmospheric Chemistry and Physics, 2020_

## Referee Comment (RC1) · Anonymous Referee #1 · 26 May 2020

The paper "Rate coefficients for reactions of OH with aromatic and aliphatic volatile organic compounds determined by the Multivariate Relative Rate Technique Âż by J. Shaw et al. describes the application of a recently developed technique for the measurement of relative rate coefficients of complex mixtures. The technique had been already presented in an earlier work by the same authors, but has been improved in this work to allow also the measurement of rate constants for slower reacting hydrocarbons. The technique has been validated by the measurement of well-known rate constants, and the overall measurements are used to validate SAR models. The rate

constants of 5 species have also been measured for the first time. The work is well done and within the scope of ACP. I recommend publication after considering the following minor remarks.

Introduction: page 3, line 13, please precise at first use what Cx substituted aromatics means. Page 4, line 9: change "." to "," Page 6: how can you be sure that all H is converted into rather unreactive HO2? Is there a possibility that H-atoms survive into the reaction zone and react more or less fast with the different VOCs? Page 12, last paragraph: is it possible that the fact that you do not find agreement between model and observations with increasing NO is due to the fact that you do not consider in your model reactions of RO2 with NO? Figure 1, 2 and 3: I don't understand what the dashed arrows mean? Does it just show the direction of where the data points should go, or has the lengths of the arrow (which is different- in the different figures) a meaning? How did you choose the x-values of these data points? In figure 3 the three red and two green dots are slightly displaced, while in Figure 4 they all have the same x-value. I guess this has no meaning? I would have expected the points on the dashed lines, and maybe the arrows down to the x-axis to indicate the retrieved rate constant. The current version is confusing to me, so please explain in more detail (maybe also in the text, not just in the legend).

---

## Referee Comment (RC2) · Anonymous Referee #2 · 27 May 2020

This is the reviewer report for manuscript entitled "Rate coefficients for reactions of OH with aromatic and aliphatic volatile organic compounds determined by the Multivariate Relative Rate Technique" by Jacob T. Shaw, Andrew R. Rickard, Mike J. Newland and Terry J. Dillon.

The paper describes an experimental gas-phase kinetic study of reactions of VOC with OH at 296 K. Particularly, the multivariate relative rate technique, developed by the authors in previous works, was employed here to determine thirty-five rate coefficients, five of which were determined for the first time. The results are in good agreement with

available literature values and SAR estimations, which indicates that this methodology is appropriate to determine simultaneously rate coefficients of VOC mixtures with OH. In this work also the sensitivity of this technique was improved and rate coefficients for slower reacting VOC with OH were determined.

In my opinion, this paper is well structured and comprehensive and it represents a significant contribution to the atmospheric chemistry. I support publishing the paper after minor corrections suggested below.

In Section 3.1, Figure S2 is mentioned in the text but Figure S1 is never mentioned in the manuscript. Maybe Figure S1 should be mention in the text of Section 2 (Methodology). Page 8, line 17, Figure S2 is related with Mixture 2 but caption of Figure S2 mentions Mixture 1, it should be checked. Figure S4 of the supplemental material (refered to Mixture 3) is never mantioned in the manuscript. Table 2 caption should be checked because it mentions Figure S3 for Mixture 2, but Figure S3 caption mentions Mixture 1.

[Figure]

---

## Referee Comment (RC3) · Anonymous Referee #3 · 31 May 2020

The manuscript entitled "Rate coefficients for reactions of OH with aromatic and aliphatic volatile organic compounds determined by the Multivariate Relative Rate Technique" by Shaw et al. reported the rate coefficients for the 35 VOCs with OH radicals (five of which are measured for the first time) at room temperature by using the recently developed multivariate relative rate technique. The obtained rate coefficients are in reasonable agreement with previously reported values and SAR calculated ones. The paper is very clear and well explained. It should be published after incorporating the minor corrections suggested below. 1. In Page3, Line 10: What do you mean by

five aromatic compounds and eleven aromatic compounds? 2. In Page 4; Line 10: The author should cite a reference. 3. What is the rate coefficient for the reaction R2? 4. How did the authors confirm the absence of any dark reactions? 5. In Page 8, Line 30: Authors should state the percentage depletion of all the studied VOCs other than t-butyl and n-pentyl benzene. 6. In Page 9, Line8: Authors should state the name of the compounds in the running text for the benefit of the readers. 7. In Page 13, Line 20: Cite the reference for the concentration of OH radicals. 8. In Figures 2 and 3, the dashed lines are very much confusing. What does it state? Does it show the rate coefficients? Please give a detailed explanation of this. 9. The caption of Figure S3 should be corrected (mixture 1 should be replaced by mixture 2) as the authors have mentioned mixture 2 in the caption of Table 2.

---

## Referee Comment (RC4) · Anonymous Referee #4 · 3 Jun 2020

General comments:

This paper entitled "Rate coefficients for reactions of OH with aromatic and aliphatic volatile organic compounds determined by the Multivariate Relative Rate Technique" by Shaw et al. presents the measurements of 35 rate coefficients at 296 K from the reactions of aromatic and aliphatic volatile organic compounds with OH radicals, using the multivariate relative rate method. This method has been presented for the first time by the same research group in the paper of Shaw et al., ACP, 2018. The experimental work presented in this paper appears to be of very high quality, precise and well-

presented. The obtained rate coefficients are in good agreement with the evaluated literature, and well supported by SAR. Overall, the paper is well written. I recommend the publication in Atmospheric Chemistry and Physics journal. Before acceptance, I have some comments and corrections to suggest.

Specific comments:

Page 5, line 1: The sentence is too general and is not really specific to the remaining gaps that the present work aims to fill. Please rephrase accordingly.

Page 9, lines 5-6: The statement that the obtained OH rate coefficients for the ten reactions is in agreement with the those from literature within errors, is not entirely correct. The obtained kOH for n-propylbenzene is not in agreement with the evaluated literature within the indicated errors. Please correct the sentence accordingly.

Page 9, lines 31-32: From the Table 4, the comparison between the measured kOH and the evaluated literature k is not evaluated. Please add a comment.

Page 27, Figure 4: In contrast to the other plots, the plot showed in Figure 4 is characterized by a higher Y-intercept which might be statistically significative, which represents more than 10% of the maximum measured depletion factor. How do you explain this observation?

Page 11, lines 24-29, Figure 8: The use of higher concentration of O2 tends to improve the accuracy of the results, as shown in Figure 8 and stated in the manuscript, in particularly in reducing the back decomposition of the OH-aromatic adduct. Based on this observation, I wonder why the use of zero air was not considered in the flow reactor, at least, replacing N2 flowing through MFC 2 (see Figure S1), introducing zero air through the mobile injector. Is there any reason that N2 was chosen preferentially?

Technical corrections:

Page 17, line 13: "www.kinetcus.com" should be replaced by "www.kintecus.com"

[Figure]

Page 21, line 4: "OH reactivity" should be replaced by "OH reactivities".

---

## Author Response (AR1)

**Authors' response to all referees for ACP-2020-281**

The authors would like to thank all the referees for their overall very positive support for this work and their helpful and insightful comments. Below is a breakdown of all referee comments (*black italics*) with appropriate author responses (blue text).

5

**Anonymous Referee #1**

The paper "Rate coefficients for reactions of OH with aromatic and aliphatic volatile organic compounds determined by the Multivariate Relative Rate Technique Âz by J. Shaw et al. describes the application of a recently developed technique for the measurement of relative rate coefficients of complex mixtures. The

10 technique had been already presented in an earlier work by the same authors, but has been improved in this work to allow also the measurement of rate constants for slower reacting hydrocarbons. The technique has been validated by the measurement of well-known rate constants, and the overall measurements are used to validate SAR models. The rate constants of 5 species have also been measured for the first time. The work is well done and within the scope of ACP. I recommend publication after considering the following minor remarks.

**15 The authors would like to thank the referee for their supportive review.**

Introduction: page 3, line 13, please precise at first use what Cx substituted aromatics means.

Cx refers to the number of carbon atoms in the substituent groups attached to the benzene ring of an aromatic molecule. We acknowledge that the first use of this terminology on page 3 could be better clarified, therefore we have changed the text to read "monoaromatics containing four carbons as part of alkyl

20 substituents (henceforth C4 aromatics)" and later further clarified on line 27 "Monoaromatics with two or three alkyl substituents, each containing one, two or three carbons (henceforth C1, C2 or C3 substituents) have relatively large POCP values meaning that they are amongst the emitted species with the greatest potential for producing ozone (Derwent et al., 2007)."

Page 4, line 9: change "." to ","

25 We are unsure which punctuation is referred to here. The only full stop (period) in Page 4, Line 9 comes at the end of a sentence which is not linked to the following sentence. We would welcome any advice / clarification from the reviewer or the editorial team on this point.

Page 6: how can you be sure that all H is converted into rather unreactive HO2? Is there a possibility that Hatoms survive into the reaction zone and react more or less fast with the different VOCs?

30 Atomic H will react particularly fast with O2 under these conditions. Reaction R2 (H + O2 + M) has a preferred rate coefficient of  $1 \times 10^{-12}$  cm3 molecule-1 s-1 at 1 bar N2 and 300 K (Atkinson et al., 2006). Given an O2 concentration of between  $10^{17}$  and  $10^{18}$  molecules cm-3, the lifetime of atomic H with respect to reaction with O2 is less than 10 µs. Any atomic H which doesn't react with O2 would be more likely to react with any NO2 ( $k = 10^{-12}$  cm-3 s = 10^{-12} cm-3 s = 10^{-12} cm-3 s = 10^{-12} s = 10^{-12} cm-3 s = 10^{-12} s = 10^{-12} cm-3 s = 10^{-12} s = 10^{-12} s = 10^{-12} cm-3 s = 10^{-12} s

 $1.47 \times 10^{-10}$  cm3 molecule-1 s-1; Su et al., 2002), or with HO2 ( $k = 8 \times 10^{-11}$  cm3 molecule-1 s-1; Atkinson et al., 2006), than with the slow reacting aromatic VOC.

Atkinson, R., Baulch, D. L., Cox, R. A., Crowley, J. N., Hampson, R. F., Hynes, R. G., Jenkin, M. E., Rossi, M. J. and Troe, J.: Evaluated kinetic and photochemical data for atmospheric chemistry: Volume II – gas phase
reactions of organic species, Atmos. Chem. Phys., 6, 3265-4055, https://doi.org/10.5194/acp-6-3625-2006, 2006. Also at http://iupac.pole-ether.fr/index.html.

Su, M.-C., Kumaran, S. S., Lim, K. P., Michael, J. V., Wagner, A. F., Harding, L. B. and Fang, D.-C.: Rate constants  $1100 \le T \le 2000$  K, for H + NO2 -> OH + NO using two shock tube techniques: Comparison of theory to experiment, J. Phys. A., 106, 8261-8270, https://doi.org/10.1021/jp0141023, 2002.

10 Page 12, last paragraph: is it possible that the fact that you do not find agreement between model and observations with increasing NO is due to the fact that you do not consider in your model reactions of RO2 with NO?

 $RO_2$  reacts with NO with a rate coefficient between approximately  $10^{-12}$  and  $10^{-11}$  cm3 molecule-1 s-1 at room temperature. This would make the  $RO_2$  + NO reaction competitive with the reaction between  $HO_2$  and NO

- 15  $(k = 8.5 \times 10^{-12} \text{ cm}^3 \text{ molecule}^{-1} \text{ s}^{-1})$  under the experimental conditions. The presence of RO2 could therefore limit the availability of NO for scavenging HO2, thereby reducing the amount of VOC reacting with OH. As the simulated results showed a drop in OH + VOC relative to experiments, the addition of RO2 to the simulations would presumably further exacerbate this. This ignores the fact that RO2 is only present after initial OH + aromatic reactions, by which time much of the NO would already have been consumed by reaction with HO2.
- 20 The reviewer is correct, however, in that the development of a more comprehensive model, incorporating RO2 and more extensive secondary chemistry, would be beneficial for elucidation, and further interpretation, of the experimental results and for model reliability.

Figure 1, 2 and 3: I don't understand what the dashed arrows mean? Does it just show the direction of where the data points should go, or has the lengths of the arrow (which is different- in the different figures) a meaning?

- 25 How did you choose the x-values of these data points? In figure 3 the three red and two green dots are slightly displaced, while in Figure 4 they all have the same x-value. I guess this has no meaning? I would have expected the points on the dashed lines, and maybe the arrows down to the x-axis to indicate the retrieved rate constant. The current version is confusing to me, so please explain in more detail (maybe also in the text, not just in the legend).
- 30 The authors agree that this is confusing to readers. The inclusion of these data points (for previously unmeasured VOCs) is to illustrate their measured depletion values (y-axis value) relative to the other compounds in the mixture. Their x-values in the figures have no meaning as the x-axis is specifically for literature rate coefficient value (which these compounds do not have). The dashed arrows are there to indicate the points at which these compounds would fall on the *OHexp* curve. The caption for Figure 2 has been updated
- 35 to clarify this and now reads "Data for *n*-pentylbenzene (A) was not used in the calculation of *OH*exp but was included on the plot to illustrate its measured depletion factor." The captions for Figure 3 and Figure 4 were

also updated accordingly. The final sentence for the captions for all three plots already provided an explanation for the use of the dashed arrows. The sentence (which read "The dashed arrows indicate that the k values for the target compounds are determined by where they would fall on the measured  $OH_{exp}$  curve.") has been moved to precede the lists of compounds and follow the sentence amended above.

**5 Anonymous Referee #2**

This is the reviewer report for manuscript entitled "Rate coefficients for reactions of OH with aromatic and aliphatic volatile organic compounds determined by the Multivariate Relative Rate Technique" by Jacob T. Shaw, Andrew R. Rickard, Mike J. Newland and Terry J. Dillon. The paper describes an experimental gas-phase kinetic study of reactions of VOC with OH at 296 K. Particularly, the multivariate relative rate technique, developed by

- 10 the authors in previous works, was employed here to determine thirty-five rate coefficients, five of which were determined for the first time. The results are in good agreement with available literature values and SAR estimations, which indicates that this methodology is appropriate to determine simultaneously rate coefficients of VOC mixtures with OH. In this work also the sensitivity of this technique was improved and rate coefficients for slower reacting VOC with OH were determined. In my opinion, this paper is well structured and
- 15 comprehensive and it represents a significant contribution to the atmospheric chemistry. I support publishing the paper after minor corrections suggested below.

The authors would like to thank the referee for their supportive review.

In Section 3.1, Figure S2 is mentioned in the text but Figure S1 is never mentioned in the manuscript. Maybe Figure S1 should be mention in the text of Section 2 (Methodology).

20 Thank you for pointing this out. The text in Section 2 has been amended to refer to Figure S1 at the end of the second sentence (Page 6, Line 4).

*Page 8, line 17, Figure S2 is related with Mixture 2 but caption of Figure S2 mentions Mixture 1, it should be checked.*

The reviewer is correct, Figure S2 should refer to Mixture 2. The text for the caption of Figure S2 has been corrected. The caption for Figure S3 also had the same error and has been corrected.

Figure S4 of the supplemental material (refered to Mixture 3) is never mantioned in the manuscript.

A reference to Figure S4 has been added to the caption for Table 1.

Table 2 caption should be checked because it mentions Figure S3 for Mixture 2, but Figure S3 caption mentions Mixture 1.

30 The caption for Figure S3 has been amended to correctly refer to Mixture 2.

**Anonymous Referee #3**

The manuscript entitled "Rate coefficients for reactions of OH with aromatic and aliphatic volatile organic compounds determined by the Multivariate Relative Rate Technique" by Shaw et al. reported the rate

coefficients for the 35 VOCs with OH radicals (five of which are measured for the first time) at room temperature by using the recently developed multivariate relative rate technique. The obtained rate coefficients are in reasonable agreement with previously reported values and SAR calculated ones. The paper is very clear and well explained. It should be published after incorporating the minor corrections suggested below.

**5 The authors would like to thank the referee for their supportive review.**

**In Page3, Line 10: What do you mean by five aromatic compounds and eleven aromatic compounds?**

The reviewer is referring to a section of the main document which currently reads, "The total concentration of five aromatic compounds was measured to be 27 (± 8) ppbv in Mumbai (Pandit et al., 2011) while summed concentrations of eleven aromatic compounds were measured to be 30 ppbv in Yokohama City

10 (Tiwari et al., 2010)." This was intended to provide some indication of the prevalence of aromatic compounds in different polluted cities. Comparing values for 5 compounds with 11 compounds, as implied by the sentence, isn't particularly valid. The sentence has therefore been split in two and rearranged to remove the implied comparison.

**In Page 4; Line 10: The author should cite a reference.**

15 The authors are unsure which sentence the reviewer is referring to. The sentence on Page 4, Line 10 reads, "In laboratory studies performed at low O2 concentrations, the competition of these pathways can lead to non-atmospherically relevant product distribution and must be considered (Ji et al., 2017; Newland et al., 2017)" and has two references. The preceding sentence doesn't have a reference but refers to two numbers in the previous sentence, which are referenced.

**20 What is the rate coefficient for the reaction R2?**

Reaction R2 (H +  $O_2$  + M) has a preferred rate coefficient of  $1.0 \times 10^{-12}$  cm3 molecule-1 s-1 at 1 bar  $N_2$  and 300 K (Atkinson et al., 2006). A reference to this rate coefficient has been added to the manuscript.

Atkinson, R., Baulch, D. L., Cox, R. A., Crowley, J. N., Hampson, R. F., Hynes, R. G., Jenkin, M. E., Rossi, M. J. and Troe, J.: Evaluated kinetic and photochemical data for atmospheric chemistry: Volume II – gas phase reactions of organic species, Atmos. Chem. Phys., 6, 3265-4055, https://doi.org/10.5194/acp-6-3625-2006,

2006. Also at http://iupac.pole-ether.fr/index.html.

**How did the authors confirm the absence of any dark reactions?**

During configuration of the experimental setup, various experiments were performed to assess the extent of VOC losses to other reactions. "Dark reactions" were tested for by comparing the measured VOC concentrations (GC-MS) for mixtures with and without H2O in the reactor whilst the lamp was removed. Further tests were conducted to compare the concentrations of VOCs with and without the reactor (i.e. through ¼" stainless steel tubing). The results of these tests showed little variation in VOC concentrations that were well within the instrumentation noise. Regardless, the presence of any reactions that occurred during "dark"

off" i.e. if they were always present. Readers are referred to Shaw et al. (2018) and references therein (e.g. Cryer et al. (2016)) at the beginning of the methodology section.

Cryer, D. R.: Measurements of hydroxyl radical reactivity and formaldehyde in the atmosphere, PhD thesis, University of Leeds, 2016.

- 5 Shaw, J. T., Lidster, R. T., Cryer, D. R., Ramirez, N., Whiting, F. C., Boustead, G. A., Whalley, L. K., Ingham, T., Rickard, A. R., Dunmore, R. E., Heard, D. E., Lewis, A. C., Carpenter, L. J., Hamilton, J. F. and Dillon, T. J.: A selfconsistent, multivariate method for the determination of gas-phase rate coefficients, applied to reactions of atmospheric VOCs and the hydroxyl radical, Atmos. Chem. Phys., 18, 4039–4054, https://doi.org/10.5194/acp-18-4039-2018, 2018b.
- 10 In Page 8, Line 30: Authors should state the percentage depletion of all the studied VOCs other than t-butyl and n-pentyl benzene.

The authors believe that a list of percentage depletions for 11 VOCs would be difficult to read within the text of the main manuscript. Instead, Table S1, comparing percentage depletions for each VOC in Mixture 2 under different NO conditions, has been added to the Supplementary Information. This table is referred to in on Page 8 Line 22

15 Page 8 Line 32.

In Page 9, Line8: Authors should state the name of the compounds in the running text for the benefit of the readers.

The sentence has been amended to refer readers to Table 3, which contains a list of all compounds in Mixture 3.

20 In Page 13, Line 20: Cite the reference for the concentration of OH radicals.

Page 13, Line 20 has been amended to include a reference to Li et al. (2018): "Atmospheric lifetimes for the eight VOCs for which rate coefficients for reaction with OH were measured for the first time, were calculated using a global 24-hour average atmospheric tropospheric concentration of  $[OH] = 1 \times 10^6$  molecules cm-3 (Li et al., 2018)."

25 Li, M., Karu, E., Brenninkmeijer, C., Fischer, H., Lelieveld, J. and Williams, J.: Tropospheric OH and stratospheric OH and Cl concentrations determined from CH4, CH3Cl, and SF6 measurements, npj Clim. Atmos. Sci., 1, 29, https://doi.org/10.1038/s41612-018-0041-9, 2018.

*In Figures 2 and 3, the dashed lines are very much confusing. What does it state? Does it show the rate coefficients? Please give a detailed explanation of this.*

30 Please see the authors' response to comments from Referee #1 above. Hopefully the updated Figure captions provide a more obvious explanation as to the meaning of these arrows.

The caption of Figure S3 should be corrected (mixture 1 should be replaced by mixture 2) as the authors have mentioned mixture 2 in the caption of Table 2.

**Anonymous Referee #4**

This paper entitled "Rate coefficients for reactions of OH with aromatic and aliphatic volatile organic compounds determined by the Multivariate Relative Rate Technique" by Shaw et al. presents the measurements of 35 rate

- 5 coefficients at 296 K from the reactions of aromatic and aliphatic volatile organic compounds with OH radicals, using the multivariate relative rate method. This method has been presented for the first time by the same research group in the paper of Shaw et al., ACP, 2018. The experimental work presented in this paper appears to be of very high quality, precise and well presented. The obtained rate coefficients are in good agreement with the evaluated literature, and well supported by SAR. Overall, the paper is well written. I recommend the
- 10 publication in Atmospheric Chemistry and Physics journal.

The authors would like to thank the referee for their supportive review.

Page 5, line 1: The sentence is too general and is not really specific to the remaining gaps that the present work aims to fill. Please rephrase accordingly

We agree that this sentence is quite generic and doesn't fit the context of the paragraph. The sentence 15 has been removed from the manuscript.

Page 9, lines 5-6: The statement that the obtained OH rate coefficients for the ten reactions is in agreement with the those from literature within errors, is not entirely correct. The obtained kOH for n-propylbenzene is not in agreement with the evaluated literature within the indicated errors. Please correct the sentence accordingly.

The reviewer is correct that the result for *n*-propylbenzene does not agree with the literature value (many thanks for pointing this error out). The sentence has been amended to read: "Results for all ten reactions, except that for *n*-propylbenzene, were in agreement with evaluated literature rate coefficients, within errors."

*Page 9, lines 31-32: From the Table 4, the comparison between the measured kOH and the evaluated literature k is not evaluated. Please add a comment.*

25 This paragraph has been reorganised slightly and an additional sentence added to compare measured  $k_{OH}$  with literature  $k_{OH}$ . The paragraph now reads:

"A fourth synthetic gas mixture, containing aliphatic VOCs with a range of sizes, branching and OH reaction rate coefficients, was also measured using the multivariate relative rate method. Figure 4 shows the relative rate plot for this VOC mixture. A clear linear relationship was observed between depletion factor and

30 literature rate coefficient. Table 4 provides a list of compounds in this mixture, along with their measured rate coefficient and evaluated literature reference rate coefficient, where available. Results for all seven of the reference reactions were in excellent agreement with evaluated literature rate coefficients, within errors. Rate coefficients for the reactions of OH with three VOCs (2-methylheptane, 2-methylnonane and ethylcyclohexane), were measured for the first time."

Page 27, Figure 4: In contrast to the other plots, the plot showed in Figure 4 is characterized by a higher Yintercept which might be statistically significative, which represents more than 10% of the maximum measured depletion factor. How do you explain this observation?

Equation 1 (also below) indicates that, when plotting depletion factor (ln[VOC]0/ln[VOC]) against kOH,
the intercept should be zero. Experimental uncertainty means that a non-zero intercept can be acceptable, but the reviewer is correct in asserting that the intercept in Figure 4 may be non-trivial.

A non-linear (curved) relationship between depletion factor and  $k_{OH}$  has previously been observed, and was reported in work introducing this technique (Shaw et al., 2018). This relationship was observed to occur for VOC mixtures with a large range in VOC + OH rate coefficients. Poor mixing within the reactor was though to 10 result in an inconsistent concentration of OH being available for reaction with VOCs, leading to the observed

non-linearity. This hypothesis was successfully replicated using kinetic models.

It could be the case that such a relationship occurred for this mixture. With no measurements of VOCs with a  $k_{OH}$  value less than  $3.8 \times 10^{-12}$  cm3 molecule-1 s-1, the relationship below this point is impossible to demonstrate and cannot reliably be extrapolated. In any case, the authors maintain that so long as a consistent

15 function can be plotted through the data to model the depletion factor vs. *k*OH curve, the actual nature of the relationship is not important when deriving rate coefficients. Extrapolation of the curve should be avoided where possible by using reference reactions which encapsulate the targeted reactions.

$$\ln\left(\frac{[\text{VOC}]_0}{[\text{VOC}]}\right) = k_{\text{VOC+OH}} \int [\text{OH}]_t dt$$

25

For more discussion on the non-linearity of these relative rate plots, the reviewers (and readers) are referred to Shaw et al. (2018), and its Supplement and Discussion.

Shaw, J. T., Lidster, R. T., Cryer, D. R., Ramirez, N., Whiting, F. C., Boustead, G. A., Whalley, L. K., Ingham, T., Rickard, A. R., Dunmore, R. E., Heard, D. E., Lewis, A. C., Carpenter, L. J., Hamilton, J. F. and Dillon, T. J.: A self-consistent, multivariate method for the determination of gas-phase rate coefficients, applied to reactions of atmospheric VOCs and the hydroxyl radical, Atmos. Chem. Phys., 18, 4039–4054, https://doi.org/10.5194/acp-18-4039-2018. 2018.

Page 11, lines 24-29, Figure 8: The use of higher concentration of O2 tends to improve the accuracy of the results, as shown in Figure 8 and stated in the manuscript, in particularly in reducing the back decomposition of the OH-aromatic adduct. Based on this observation, I wonder why the use of zero air was not considered in the flow reactor, at least, replacing N2 flowing through MFC 2 (see Figure S1), introducing zero air through the

30 mobile injector. Is there any reason that N2 was chosen preferentially?

High purity (grade 6, 99.9999% pure)  $N_2$  was initially chosen to reduce contamination and the potential for further reactive chemistry after the initial oxidation of the primary VOCs. The back decomposition of the intermediate OH-adduct only occurs for aromatic VOCs and not for aliphatic VOCs such as alkanes and alkenes. The high purity  $N_2$  was carried over from the previous primarily aliphatic experiments (Shaw et al., 2018) but the

reviewer is correct in that experiments could be repeated with zero air to better quantify the impact of higher concentrations of  $O_2$ .

Shaw, J. T., Lidster, R. T., Cryer, D. R., Ramirez, N., Whiting, F. C., Boustead, G. A., Whalley, L. K., Ingham, T., Rickard, A. R., Dunmore, R. E., Heard, D. E., Lewis, A. C., Carpenter, L. J., Hamilton, J. F. and Dillon, T. J.: A self-

5 consistent, multivariate method for the determination of gas-phase rate coefficients, applied to reactions of atmospheric VOCs and the hydroxyl radical, Atmos. Chem. Phys., 18, 4039–4054, https://doi.org/10.5194/acp-18-4039-2018, 2018.

Page 17, line 13: "www.kinetcus.com" should be replaced by www.kintecus.com

Thank you for pointing this out! This has been corrected.

10 Page 21, line 4: "OH reactivity" should be replaced by "OH reactivities".

This has been corrected.

[revised manuscript text omitted]
 8$                                                                                                  | $79\pm20$                                                                                                       | Atkinson and Arey, 2003 | 10                                      |
| 1,2,3,5-tetramethylbenzene | $62\pm9$                                                                                                   | $62.4\pm0.8$                                                                                                    | Alarcón et al., 2015    | 1                                       |
| 1,3,5-trimethylbenzene     | $60\pm5$                                                                                                   | $57 \pm 11$                                                                                                     | Calvert et al., 2002    | 5                                       |
| 1,2,4,5-tetramethylbenzene | $59\pm12$                                                                                                  | $55.5\pm3.4$                                                                                                    | Aschmann et al., 2013   | 2                                       |
| α-pinene                   | $53\pm8$                                                                                                   | 53 (+22)                                                                                                        | Atkinson et al., 2006   | 9                                       |
| 1,2,4-trimethylbenzene     | $34 \pm 3$                                                                                                 | $33 \pm 8$                                                                                                      | Calvert et al., 2002    | 5                                       |
| 1,2,3-trimethylbenzene     | $38 \pm 4$                                                                                                 | $33 \pm 8$                                                                                                      | Calvert et al., 2002    | 5                                       |
| m -xylene           | $21 \pm 3$                                                                                                 | $23 \pm 3$                                                                                                      | Calvert et al., 2002    | 15                                      |
| o-xylene                   | $10 \pm 4$                                                                                                 | $13 \pm 3$                                                                                                      | Calvert et al., 2002    | 10                                      |

Table 2: Results of relative rate experiments, with evaluated literature data, for each VOC in Mixture 2, ordered by evaluated literature k value. The results of experiments using different concentrations of NO are presented, along with a mean measured k value for each VOC, weighted to the errors in the individual NO experiments. Each iteration with a different [NO] comprised three individual experiments with different OH reactivities of mixture injected into the reactor. Figure S3 shows the relative rate plots for Mixture 2 with an OH reactivity (in the reactor) of 18 s-1 and NO concentrations of 0, 30 and 70 ppb.

|                         | Measured k (296 K) / 10 -12 cm 3 molecule -1 s -1 |                                                |                                                | molecule -1 s -1 | Evaluated literature                                                                    |                         | Number of                  |
|-------------------------|-----------------------------------------------------------------------------------------------|------------------------------------------------|------------------------------------------------|----------------------------------------|-----------------------------------------------------------------------------------------|-------------------------|----------------------------|
| Name                    | [NO] =
0 ppb                                                                               | [NO] =
30 ppb                               | [NO] =
70 ppb                               | Weighted
average                    | k (298 K) / 10 -12 cm 3
molecule -1 s -1 | Reference               | literature
measurements |
| m -xylene        | $\begin{array}{c} 20.6 \pm \\ 0.7 \end{array}$                                                | $\begin{array}{c} 19.6 \pm \\ 0.4 \end{array}$ | 19.8 ±
1.3                                  | $19.9\pm0.5$                           | $23 \pm 3$                                                                              | Calvert et al. 2002     | 15                         |
| 3-ethyltoluene          | $\begin{array}{c} 19.7 \pm \\ 0.9 \end{array}$                                                | $\begin{array}{c} 21.7 \pm \\ 0.4 \end{array}$ | 19.4 ± 1.0                                     | $21.1\pm1.2$                           | $19\pm7$                                                                                | Calvert et al. 2002     | 2                          |
| o-xylene                | 12.3 ± 1.3                                                                                    | $\begin{array}{c} 12.4 \pm \\ 0.4 \end{array}$ | $\begin{array}{c} 12.3 \pm \\ 0.6 \end{array}$ | $12.4\pm0.1$                           | 13 ± 3                                                                                  | Calvert et al. 2002     | 10                         |
| 4-ethyltoluene          | $14 \pm 2$                                                                                    | $\begin{array}{c} 14.8 \pm \\ 0.5 \end{array}$ | 14.1 ±
1.2                                  | $14.6\pm0.3$                           | $12 \pm 4$                                                                              | Calvert et al. 2002     | 2                          |
| 2-ethyltoluene          | 13.4 ± 1.0                                                                                    | 15.0±
1.1                                   | $\begin{array}{c} 12.7 \pm \\ 0.1 \end{array}$ | $12.7\pm0.2$                           | $12 \pm 4$                                                                              | Calvert et al. 2002     | 2                          |
| Ethylbenzene            | $5\pm 2$                                                                                      | 6.0 ±
0.3                                   | 6.7 ±
0.5                                   | $6.1 \pm 0.4$                          | $7.0\pm2$                                                                               | Calvert et al. 2002     | 3                          |
| Isopropylbenzene        | $\begin{array}{c} 6.5 \pm \\ 0.5 \end{array}$                                                 | 6.4 ±
0.4                                   | $\begin{array}{c} 6.5 \pm \\ 0.6 \end{array}$  | $6.45\pm0.02$                          | $6.3 \pm 2$                                                                             | Calvert et al. 2002     | 3                          |
| n -propylbenzene | $\begin{array}{c} 8.9 \pm \\ 0.7 \end{array}$                                                 | $\begin{array}{c} 8.7 \pm \\ 0.6 \end{array}$  | $7\pm2$                                        | $8.7\pm0.5$                            | $5.8 \pm 1.5$                                                                           | Calvert et al. 2002     | 3                          |
| Toluene                 | $5\pm 2$                                                                                      | $0\pm 2$                                       | $7\pm2$                                        | $4 \pm 4$                              | 5.6 (+1:2)                                                                              | Atkinson et al.
2006 | 18                         |
| t -butylbenzene  | 3.5 ±
1.1                                                                                  | 3.4 ± 0.5                                      | 3.1 ± 0.7                                      | $3.3 \pm 0.1$                          | $4.5\pm2$                                                                               | Calvert et al. 2002     | 2                          |
| n -pentylbenzene | $3\pm 5$                                                                                      | $\begin{array}{c} 5.4 \pm \\ 0.4 \end{array}$  | $1\pm4$                                        | $5.3\pm0.7$                            |                                                                                         |                         |                            |

| Name                    | Measured k (296
K) / 10 -12 cm 3
molecule -1 s -1 | Evaluated literature
k (298 K) / 10 -12
cm- molecule -1 s -1 | Reference             | Number of
literature
measurements |
|-------------------------|------------------------------------------------------------------------------------------------------------|-----------------------------------------------------------------------------------------------------|-----------------------|-----------------------------------------|
| o-xylene                | $13.0\pm1.2$                                                                                               | $13 \pm 3$                                                                                          | Calvert et al. 2002   | 10                                      |
| 2-ethyltoluene          | $12.2\pm0.8$                                                                                               | $12 \pm 4$                                                                                          | Calvert et al. 2002   | 2                                       |
| n -decane        | $10.7\pm0.7$                                                                                               | $11 \pm 2$                                                                                          | Atkinson, 2003        | 6                                       |
| n -nonane        | $10.1\pm1.2$                                                                                               | $10 \pm 2$                                                                                          | Atkinson, 2003        | 9                                       |
| Ethylbenzene            | $6.6\pm0.5$                                                                                                | $7\pm2$                                                                                             | Calvert et al. 2002   | 3                                       |
| n -propylbenzene | $6.8\pm0.3$                                                                                                | $5.8 \pm 1.5$                                                                                       | Atkinson, 2003        | 3                                       |
| Toluene                 | $4\pm 2$                                                                                                   | 5.6 (+1.5)                                                                                          | Atkinson et al., 2006 | 18                                      |
| 2,2,3-trimethylbutane   | $6\pm 6$                                                                                                   | $3.8 \pm 1.0$                                                                                       | Atkinson, 2003        | 6                                       |
| 1,3-diethylbenzene      | $22 \pm 4$                                                                                                 |                                                                                                     |                       |                                         |
| 1,4-diethylbenzene      | $16 \pm 4$                                                                                                 |                                                                                                     |                       |                                         |
| 1,2-diethylbenzene      | $14 \pm 4$                                                                                                 |                                                                                                     |                       |                                         |
| n -butylbenzene  | $11 \pm 4$                                                                                                 |                                                                                                     |                       |                                         |
| n -pentylbenzene | $7\pm2$                                                                                                    |                                                                                                     |                       |                                         |

Table 3: Results of relative rate experiments, with evaluated literature data, for each VOC in Mixture 3, ordered by evaluated literature *k* value. The measured values presented here are the weighted averages of five individual experiments conducted with different mixture OH reactivities and 40 ppbv NO. Five VOCs had no reported data for their reactions with OH.

| Name                  | Measured k (295
K) / 10 -12 cm 3
molecule -1 s -1 | Evaluated literature k
(298 K) / 10 -12 cm 3
molecule -1 s -1 | Reference      | Number of
literature
measurements |
|-----------------------|------------------------------------------------------------------------------------------------------------|-----------------------------------------------------------------------------------------------------------------|----------------|-----------------------------------------|
| Cyclooctane           | $13.7\pm0.3$                                                                                               | $13 \pm 7$                                                                                                      | Atkinson, 2003 | 2                                       |
| n -undecane    | $12.3\pm0.3$                                                                                               | $12 \pm 2$                                                                                                      | Atkinson, 2003 | 2                                       |
| Cycloheptane          | $11.4\pm0.3$                                                                                               | $12 \pm 3$                                                                                                      | Atkinson, 2003 | 3                                       |
| n -decane      | $10.3\pm0.3$                                                                                               | $11 \pm 2$                                                                                                      | Atkinson, 2003 | 6                                       |
| n -nonane      | $11.0\pm0.3$                                                                                               | $10 \pm 2$                                                                                                      | Atkinson, 2003 | 9                                       |
| n -octane      | $8.8\pm0.3$                                                                                                | $8\pm 2$                                                                                                        | Atkinson, 2003 | 6                                       |
| 2-methylpentane       | $4.6 \pm 0.3$                                                                                              | $5.2 \pm 1.3$                                                                                                   | Atkinson, 2003 | 4                                       |
| 3-methylpentane       | $5.3 \pm 0.3$                                                                                              | $5.2 \pm 1.3$                                                                                                   | Atkinson, 2003 | 3                                       |
| 2,2,3-trimethylbutane | $3.9\pm0.3$                                                                                                | $3.8 \pm 1.0$                                                                                                   | Atkinson, 2003 | 6                                       |
| 2-methylheptane       | $9.1 \pm 0.3$                                                                                              |                                                                                                                 |                | 0*                                      |
| 2-methylnonane        | $11.0\pm0.3$                                                                                               |                                                                                                                 |                | 0                                       |
| Ethylcyclohexane      | $14.4\pm0.3$                                                                                               |                                                                                                                 |                | 0*                                      |

Table 4: Results of relative rate experiments, with evaluated literature data, for each VOC in Mixture 4, ordered by evaluated literature *k* value.

\* Shaw et al. (2018) reported *k*OH(323 K) for these compounds butand no literature *k*OH value was found at 298 K.

**Figures**

Figure 1: Relative rate plot for Mixture 1 (OH reactivity of 48 s-1) in the absence of NO at 296 ( $\pm$  2) K. Compounds with a reference rate coefficient for reaction with OH using evaluated literature values. Error bars on the *y*-axis, equal to 1 standard error, were calculated by combining the standard error in peak areas for eight lamp-off and eight lamp-on samples. Error bars on the *x*-axis were typically large (approximately  $\pm$  20-30 %) and accounted for deviations from the predicted trend for all VOCs. A weighted (to the uncertainty in the y values) linear fit was used to generate the slope with *OHexp* = 1.8 ( $\pm$  0.1) × 109 molecules cm-3 s and R2 = 0.980. The VOCs can be identified as follows: 1, *o*-xylene; 2, *m*-xylene; 3, 1,2,4-trimethylbenzene; 4, 1,2,3-trimethylbenzene; 5, *a*-pinene; 6, 1,2,4,5-tetramethylbenzene; 7, 1,3,5-trimethylbenzene; 8, 1,2,3,5-tetramethylbenzene; 9, β-pinene; 10, isoprene.

10